# Near-Optimal Consistency-Robustness Trade-Offs
# for Learning-Augmented Online Knapsack Problems

**Mohammadreza Daneshvaramoli** [1]   **Helia Karisani** [1]   **Adam Lechowicz** [1]   **Bo Sun** [2]   **Cameron Musco** [1]
**Mohammad Hajiesmaili** [1]

## Abstract

This paper introduces a family of learning-augmented algorithms for online knapsack problems that achieve near Pareto-optimal consistency-robustness trade-offs through a simple combination of trusted learning-augmented and worst-case algorithms. Our approach relies on succinct, practical predictions—single values or intervals estimating the minimum value of any item in an offline solution. Additionally, we propose a novel fractional-to-integral conversion procedure, offering new insights for online algorithm design.

## 1. Introduction

Learning-augmented design (Lykouris & Vassilvtiskii, 2018; Purohit et al., 2018) is a successful framework whose goal is to systematically go beyond competitive analysis in online problems by leveraging ML predictions to improve algorithm performance without sacrificing worst-case guarantees. In this framework, online algorithms are evaluated using the concepts of *consistency* and *robustness*, which characterize the *competitive ratio* (see § 2.1) when the prediction is accurate or arbitrarily wrong, respectively. An overarching design goal is to achieve a Pareto-optimal result that captures the best achievable trade-off between consistency and robustness. Toward this goal, algorithms have been proposed that achieve optimal consistency-robustness trade-offs for several problems such as ski rental (Bamas et al., 2020; Wei & Zhang, 2020), online search (Sun et al., 2021a), and bin packing (Angelopoulos et al., 2024). However, Pareto-optimal results remain open for several other online problems, including the online knapsack problem.

[1]Manning College of Information and Computer Sciences, University of Massachusetts Amherst, Amherst, MA, USA [2]Cheriton School of Computer Science, University of Waterloo, Waterloo, ON, Canada. Correspondence to: Mohammadreza Daneshvaramoli <mdaneshvaram@umass.edu>, Helia Karisani <hkarisani@umass.edu>.

*Proceedings of the $42^{nd}$ International Conference on Machine Learning*, Vancouver, Canada. PMLR 267, 2025. Copyright 2025 by the author(s).

In the online knapsack problem (OKP), the goal is to pack a finite number of sequentially arriving items with different values and weights into a knapsack with limited capacity, so that the total value of admitted items is maximized. In the online setting, the decision maker must immediately and irrevocably admit or reject an item upon its arrival without knowing future items. OKP captures a broad range of resource allocation problems with applications to online advertising (Zhou et al., 2008), resource allocation (Zhang et al., 2017; Buchbinder & Naor, 2009), dynamic pricing (Bostandoost et al., 2023), supply chain (Ma et al., 2019), and beyond. Historically, work on OKP variants has focused on developing algorithms under the framework of competitive analysis (Borodin et al., 1992). It is well known that no online algorithm can achieve bounded competitive ratio for the basic version of OKP (Marchetti-Spaccamela & Vercellis, 1995; Böckenhauer et al., 2014b) without making additional assumptions on the input, e.g., assuming that value-to-weight ratios are bounded (Zhou et al., 2008; Sun et al., 2021a; Yang et al., 2021). In this work, we go beyond competitive analysis, to study OKP through the lens of learning-augmented design.

**Prior results.** There has been substantial work on learning-augmented algorithms for OKP. We classify them into three categories: (i) OKP with frequency predictions (Im et al., 2021), (ii) advice complexity of OKP (Böckenhauer et al., 2014b), and (iii) Pareto-optimal algorithms for simplified OKP variants (Sun et al., 2021a; Lee et al., 2022; Balseiro et al., 2023). See § A.1 for a complete literature review. In the following, we focus on closely related prior work.

In the first category, Im et al. (2021) study learning-augmented OKP using *frequency predictions*, i.e., predictions of the total weights of items for each possible unit value in an instance. The authors provide upper bounds on the consistency and robustness of their SENTINEL algorithm, but do not provide a lower bound on the best achievable consistency-robustness trade-off under the frequency prediction model, leaving the question of optimality open. Further, the frequency prediction model requires the algorithm to be given a large number of predictions (one for each possible unit value). Due to this complexity, accurate frequency predictions may be difficult to obtain in practice.

*Table 1.* Summary of contributions.

| Prediction Model | Algorithm | Upper Bound | Lower Bound |
|---|---|---|---|
| Point Prediction | PP-b, PP-a | $2, 1 + \min\{1, \hat{\omega}\}$ (Theorems 3.2, 3.3) | $1 + \min\{1, \hat{\omega}\}$ (Theorem 3.1) |
| Interval Prediction | IPA | $2 + \ln(u/\ell)$ (Theorem 3.4) | $2 + \ln(u/\ell)$ (Theorem 3.6) |
| Consistency-robustness | MIX | $2/\lambda$-consistent ($\lambda \in (0,1)$) 
 $\frac{\ln(U/L)+1}{(1-\lambda)}$-robust (Theorem 4.1)* | $2/\lambda$-consistent ($\lambda \in (0,1)$) 
 $\Omega\left(\frac{\ln(U/L)+1}{(1-\lambda)}\right)$-robust (Theorem 4.2)* |

▶ We present our results in the context of online fractional knapsack (OFKP); in Theorem 5.2, we give a generic conversion to the integral case, showing that the results extend with small loss to this setting under standard assumptions. Our lower bounds in the fractional case also hold in the integral case by nature of the relaxation; see § 5.   ▶ $\hat{\omega}$ is the total weight of items with critical value $\hat{v}$ (see Def 2.2).   ▶ $\ell$ and $u$ are lower and upper bound predictions on the critical value $\hat{v}$ (see Def. 2.3).   ▶ $\lambda \in (0,1)$ is a *trust hyperparameter* passed to a learning-augmented algorithm that leverages untrusted predictions.   ▶ * indicates that we consider the setting where item values are bounded in $[L, U]$.

In the second category, Böckenhauer et al. (2014b) study the advice complexity of OKP, giving an online algorithm with competitive ratio $1 + \epsilon$ for any constant $\epsilon > 0$ using $O(\log n)$ bits of advice (Böckenhauer et al., 2014b, Theorems 13 and 17), where $n$ is the number of items. To achieve this result, their algorithm requires strong advice about the offline optimal solution (including the number of items, multiple critical item values, and the indices of certain admitted items). With such strong advice, the algorithm itself is simple: it reserves the capacity for items from advice and greedily admits other items. However, acquiring the required advice from an ML model in practice is likely difficult, e.g., admitted item index advice becomes invalid if instances are shuffled. Further, Böckenhauer et al. (2014b) focus only on the setting where the advice is assumed to be accurate. It is unclear how to extend their analysis to give consistency-robustness trade-offs.

In the third category, prior works have developed algorithms with Pareto-optimal consistency-robustness trade-offs for various simplified versions of OKP. For example, using similar frequency predictions to (Im et al., 2021), Balseiro et al. (2023) develop Pareto-optimal algorithms for the *single-leg revenue management problem*, which can be interpreted as an OKP variant with unit weight items that take values from a discrete and finite set. However, the algorithms and their Pareto-optimality guarantees do not extend to the case of continuous-valued items. Pareto-optimal algorithms using succinct single-valued predictions have also been given for online conversion problems (Sun et al., 2021a; Lee et al., 2022). These problems can be seen as fractional variants of OKP with item weights equal to the capacity, where the optimal offline solution simply fills the knapsack with the single most valuable item. The consistency-robustness trade-offs derived in these settings do not extend to general OKP— in online conversion and search, an accurate prediction of the highest item value is sufficient to obtain a 1-competitive algorithm. This is not the case for OKP, since the most valuable items may be small, forcing the offline optimal solution to admit less valuable items to fill the knapsack capacity.

In summary, prior work on learning-augmented algorithms for OKP fails to achieve Pareto-optimal trade-offs in all but a few significantly simplified variants of the problem. Further, existing algorithms often leverage complex and potentially impractical predictions. Motivated by these limitations, the main goal of this paper is to answer the following question: *Can we design near **Pareto-optimal** learning-augmented algorithms for OKP using practical & **succinct predictions**?*

**Contributions.** We present OKP algorithms that achieve near Pareto-optimal trade-offs between consistency and robustness, using succinct predictions of a *critical value* $\hat{v}$, i.e., the minimum value of any item admitted by an offline optimal solution (see Table 1). Our work introduces novel algorithmic frameworks, and technical contributions to knapsack problems under both trusted and untrusted predictions.

▶ For trusted point predictions of the critical value $\hat{v}$, we prove the *optimal* competitive ratio is $1 + \min\{1, \hat{\omega}\}$, where $\hat{\omega}$ is the total weight of items with value $\hat{v}$ (Theorem 3.1). Our algorithm PP-a achieves this bound using a novel *reserve-while-greedy* approach (Theorem 3.3). Existing OKP algorithms are mainly based on a threshold-based algorithm, which is a *reserve-then-greedy* strategy that pre-allocates capacity for high-value items and greedily admits others (Lee et al., 2022; Sun et al., 2022). Instead of relying on fixed reservation, PP-a dynamically adjusts its reserved capacity based on observations of high-value items, improving the competitive ratio from 2 (achieved by a reserve-then-greedy algorithm PP-b) to $1 + \min\{1, \hat{\omega}\}$.

▶ For trusted interval predictions $\ell \leq \hat{v} \leq u$, our IPA algorithm achieves a competitive ratio of $2 + \ln(u/\ell)$, matching a derived lower bound (Theorem 3.4, 3.6). Interval predictions also model point predictions with bounded error, enabling robust solutions under uncertainty.

▶ For untrusted predictions, we propose an algorithm, MIX that combines trusted algorithms with a robust baseline. Assuming unit values lie within $[L, U]$, we show MIX achieves $c/\lambda$-consistency and $\frac{\ln(U/L)+1}{1-\lambda}$-robustness (Theorem 4.1), where $c$ is the competitive ratio of the trusted prediction algorithm. This trade-off is near-optimal (Theorem 4.2).

▶ We then introduce a fractional-to-integral conversion algorithm that adapts any OFKP solution for use in OIKP under small item weights (Zhou et al., 2008; Im et al., 2021). This conversion facilitates near-optimal consistency-robustness trade-offs for integral settings (Theorem 5.2).

▶ Finally, in § 6, we evaluate our algorithms on synthetic and real data from Bitcoin and Google workload traces. Our algorithms significantly outperform baselines without

predictions, even with noisy inputs. Compared to algorithms relying on complex frequency predictions (Im et al., 2021), our succinct prediction-based approaches exhibit superior performance and degrade gracefully with prediction errors.

## 2. Problem Formulation and Preliminaries

In this section, we define the online knapsack problem (OKP), introduce the prediction models used, and review existing algorithms and results.

Throughout the paper, we occasionally use the term *value* to refer to the unit value (i.e., value-to-weight ratio) of an item, especially in early sections for simplicity. This shorthand is common in the literature on learning-augmented knapsack (Im et al., 2021; Sun et al., 2022), and we clarify the meaning when ambiguity might arise.

**Online knapsack problem.** Consider a knapsack with capacity 1 (w.l.o.g.). Items arrive online, each with a unit value $v_i$ and weight $w_i$. Upon arrival, an algorithm decides $x_i \in \mathcal{X}_i$, the acceptance of item $i$, without future knowledge. Here $\mathcal{X}_i$ denotes feasible decisions for item $i$. Each decision $x_i$ yields profit $x_i v_i$, aiming to maximize total profit under capacity constraints. When $\mathcal{X}_i = \{0, w_i\}$, the algorithm decides to pack or reject the entire item (OIKP). For $\mathcal{X}_i = [0, w_i]$, fractional packing is allowed (OFKP). We use OKP for both problems unless otherwise specified. An OKP instance $\mathcal{I} = \{(v_i, w_i)\}_{i \in [n]}$ has the offline formulation:

$$\max_{\{x_i\}_{i \in [n]}} \sum_{i=1}^{n} x_i v_i, \text{ s.t. } \sum_{i=1}^{n} x_i \leq 1, x_i \in \mathcal{X}_i : \forall i \in [n]. \quad (1)$$

The offline solution for OFKP sorts items by unit value and fills the knapsack in that order. The *critical value* is the smallest value of items (possibly fractionally) packed. Items with higher values are all packed. This algorithm also applies approximately to OIKP when $w_i \ll 1, \forall i \in [n]$. In the online setting, OFKP and OIKP with small weights can both be solved optimally on worst-case instances via a threshold-based algorithm (see § 2.1).

**Assumptions.** We adopt the standard small weight assumption for OIKP, where $w_i \ll 1, \forall i \in [n]$ (Zhou et al., 2008). This assumption is essential to achieve a meaningful competitive ratio in worst-case instances of OIKP (Marchetti-Spaccamela & Vercellis, 1995; Zhou et al., 2008). In Theorem A.1, we show that it remains necessary for OIKP even with a trusted prediction of the critical value. Notably, our OFKP algorithms do not rely on this assumption.

Prior work on OKP and related problems, such as one-way trading and online search (El-Yaniv et al., 2001), often assumes bounded support for unit values, i.e., $v_i \in [L, U]$, where $L$ and $U$ are known. This assumption is also critical for a bounded competitive ratio in worst-case instances (Marchetti-Spaccamela & Vercellis, 1995; Zhou et al., 2008). Notably, our learning-augmented algorithms

for OFKP with trusted predictions (§ 3) do not require the bounded value assumption. However, in the untrusted prediction setting (§ 4), this assumption ensures bounded robustness by using the worst-case optimal algorithm (Zhou et al., 2008) as a subroutine. The conversion algorithm linking OFKP and OIKP (§ 5) also leverages this assumption. Additionally, Theorem A.2 shows that the bounded value assumption is necessary to achieve a bounded competitive ratio, even with a trusted prediction.

**Adversary model.** Throughout this work, we adopt the standard *adaptive adversary model* for online algorithms (Borodin & El-Yaniv, 1998). In this model, the adversary can generate the input sequence adaptively and change it based on the algorithm's past decisions. This setting captures worst-case scenarios where item arrivals are influenced by the algorithm's behavior, and it is widely used in the analysis of online problems, including online knapsack (Zhou et al., 2008; Sun et al., 2022). All our lower bounds and robustness results assume the adaptive adversary model.

### 2.1. Preliminaries

**Competitive ratio.** OKP has been historically studied using competitive analysis, where the goal is to design an online algorithm that achieves a large fraction of the offline optimum profit on all inputs (Borodin et al., 1992). We denote by $\text{OPT}(\mathcal{I})$ the offline optimum for input $\mathcal{I}$, and by $\text{ALG}(\mathcal{I})$ the profit obtained by an online algorithm. The competitive ratio is then defined as $c := \max_{\mathcal{I} \in \Omega} \text{OPT}(\mathcal{I})/\text{ALG}(\mathcal{I})$, where $\Omega$ denotes the set of all possible inputs, and ALG is said to be $c$-competitive. If ALG is randomized, we analogously define the competitive ratio as $c := \max_{\mathcal{I} \in \Omega} \text{OPT}(\mathcal{I})/\mathbb{E}[\text{ALG}(\mathcal{I})]$.

**Consistency and robustness.** In the literature on learning-augmented algorithms, competitive analysis is interpreted using the notions of *consistency* and *robustness*, introduced by Lykouris & Vassilvtiskii (2018); Purohit et al. (2018). An online algorithm with predictions is said to be $b$-*consistent* if it is $b$-competitive when given a *correct prediction*, and $r$-*robust* if it is $r$-competitive when given *any prediction*, regardless of its correctness. The goal is to design an algorithm that achieves the best robustness for any chosen consistency – i.e., achieving a *Pareto-optimal trade-off* between consistency and robustness.

**Prior results: competitive algorithms without prediction.** Both OIKP with small weights and OFKP have received significant attention. Under the assumption of bounded unit values (i.e., $v_i \in [L, U], \forall i \in [n]$), prior works (Zhou et al., 2008; Sun et al., 2021b) have shown an optimal deterministic threshold-based algorithm (ZCL) for OIKP and OFKP. We present pseudocode for ZCL in the Appendix, in Alg. 4. Note that ZCL requires prior knowledge about the maximum unit value $U$ and minimum unit value $L$ for an instance,

achieving the optimal competitive ratio $\ln(U/L) + 1$ for both OFKP and OIKP. In § 3, we show that this competitive ratio can be improved to a constant with no assumptions on $L$ and $U$ by leveraging trusted (i.e., accurate) predictions. To achieve a Pareto-optimal consistency-robustness trade-off in the untrusted prediction setting, our MIX algorithm uses the worst-case optimal ZCL algorithm as a subroutine.

### 2.2. Succinct Prediction Model

Throughout the paper we consider *succinct predictions* derived from the *critical value* of the offline optimal fractional OFKP solution, defined formally below.

**Definition 2.1** (Critical valued items $(\hat{v}, \hat{\omega})$). Given an instance of OKP, $\hat{v}$ is a *critical value* denoting the minimum unit value of any item (fractionally) admitted by the offline optimal solution. Let $\hat{\omega}$ be the total weight of the items with value $\hat{v}$. Note that $\hat{v} = \min\{v_i : \sum_{j : v_j > v_i} w_j < 1\}$.

Below, we introduce two prediction models of the critical value $\hat{v}$ of an instance. We note that our algorithms do not receive a prediction of the corresponding capacity $\hat{\omega}$.

**Definition 2.2** (Prediction Model I: Point Prediction). A single value prediction of the critical value $\hat{v}$, as defined in Def. 2.1, is given to a learning-augmented online algorithm.

The point prediction is simple; however, we note that even when an algorithm is given an accurate point prediction of $\hat{v}$, it cannot necessarily reconstruct the optimal solution online. In particular, the offline optimal solution fully accepts any item with a unit value strictly greater than $\hat{v}$. However, it may only partially admit item(s) with value $\hat{v}$. Hence, even with a perfect prediction of the exact value $\hat{v}$, the online decision-maker cannot optimally solve the problem since the optimal admittance policy for items with exact value $\hat{v}$ is unclear. In practice, one may argue that obtaining a point prediction of the exact value $\hat{v}$ is almost impossible, motivating a more coarse-grained *interval prediction*.

**Definition 2.3** (Prediction Model II: Interval Prediction). A lower bound $\ell$ and upper bound $u$ on the actual value of $\hat{v}$ (Def 2.1) are given to the learning-augmented online algorithm, satisfying the condition $\ell \leq \hat{v} \leq u$.

In this prediction model, the quality of the prediction degrades as the ratio $u/\ell$ increases. In an extreme case of $u = \ell$, the interval prediction degenerates to the aforementioned point prediction. On the other hand, with $u = U$ and $\ell = L$, the problem degenerates to a classic OKP problem (i.e., without predictions but with bounded value assumptions) when only prior knowledge of the unit value bounds is available.

**Comparisons to related prediction models.** The notion of predicting the critical value has been explored in a special case of OFKP known as the online search problem (Sun et al., 2021a; Lee et al., 2024). In this problem, all item weights are set to 1, making the critical value equivalent to the maximum value, i.e., the offline optimal solution only admits one item with the maximum value. Accurate prediction of the critical value enables the recovery of the optimal solution in this special case. However, in OFKP with arbitrary item sizes, leveraging such predictions becomes notably more challenging. One main contribution of this study is the design of an optimal competitive algorithm using an accurate prediction of the critical value in OFKP.

The integral OKP has primarily been explored using other prediction models. Im et al. (2021) propose a frequency (interval) prediction model, predicting the total weights of items for each possible unit value in an instance. On the other hand, Balseiro et al. (2023) introduce a model that focuses on a frequency prediction over the possible unit values admitted by the offline optimal algorithms. However, any frequency prediction model demands a large number of predictions, significantly increasing the complexity of obtaining predictions with high accuracy. In contrast, our succinct prediction models only require a point or interval prediction, and can achieve comparable empirical performance to algorithms that employ more complicated predictions.

Our succinct prediction is strictly weaker than the frequency prediction model. Formally, let $\{(f^\ell(v), f^u(v))\}_{v \in [L,U]}$ denote the frequency prediction (as given by Im et al. (2021)), where $f^\ell(v)$ and $f^u(v)$ denote the lower and upper bounds of the frequency density, respectively. Given such a frequency prediction, we can directly obtain an interval prediction over the critical value by calculating $\ell := \min\{v \in [L,U] : \int_v^U f^\ell(v)dv \leq 1\}$ and $u := \min\{v \in [L,U] : \int_v^U f^u(v)dv \leq 1\}$. Last, we note that developing an ML model to generate predictions is beyond the scope of this paper, as our focus is on leveraging predictions to design and analyze learning-augmented algorithms.

## 3. Algorithms with Trusted Predictions

We first present lower bound results for OKP algorithms with trusted point predictions. Then, we design OFKP algorithms for trusted point and interval predictions that achieve optimal competitive ratios (i.e., matching a lower bound).

### 3.1. Lower Bound for Trusted Point Predictions

In what follows, we show that even with an exact point prediction of the critical value $\hat{v}$, no deterministic or randomized learning-augmented online algorithm can achieve 1-competitiveness (i.e., no algorithm can match the offline optimal solution on all instances).

**Theorem 3.1.** *Given an exact prediction on the critical value $\hat{v}$, no (deterministic or randomized) online algorithm for OFKP can achieve a competitive ratio smaller than $1 + \min\{1, \hat{\omega}\}$.*

Recall that $\hat{\omega}$ is the total weight of items with the critical value $\hat{v}$. This result implies a 2-competitive lower bound when $\hat{\omega} \geq 1$, even with an accurate point prediction of $\hat{v}$. To understand the intuition, consider a special case when $\hat{\omega} = 1$. If an algorithm is presented with the first item $(\hat{v}, 1)$, it cannot admit the entire item, as the next item could be $(\infty, 1 - \epsilon)$ (with $\epsilon \to 0$); on the other hand, it must admit a portion of this item since there may be no following item. The optimal balance is to admit $1/2$ of the first item, giving a 2-competitive lower bound. A refined bound can be attained for instances with smaller values of $\hat{\omega}$. The full proof is given in Appendix A.6.1. By nature of the OFKP relaxation, Theorem 3.1 also lower bounds the harder case of OIKP.

### 3.2. OFKP Algorithms with Trusted Point Predictions

We first present PP-b, a trusted-prediction algorithm for OFKP that is 2-competitive when given the exact prediction $\hat{v}$. Then, we present PP-a, a refined algorithm that improves the instance-dependent competitive ratio to $1 + \min\{1, \hat{\omega}\}$, matching the lower bound. Recall that $\hat{\omega}$ is unknown and not provided by a prediction (see Def. 2.1). We note that a naïve trusted-prediction algorithm that accepts all items with values $\geq \hat{v}$ results in an arbitrarily large worst-case competitive ratio (see Appendix A.5 and A.6.4).

**PP-b:** *A basic trusted point-prediction 2-competitive algorithm.* We present an algorithm (pseudocode in Alg. 5) that, given the exact value of $\hat{v}$, is 2-competitive for OFKP. The idea is to set aside half of the capacity for high-value items ($> \hat{v}$) and allocate the other half for *minimum acceptable* items with value $\hat{v}$. By doing so, PP-b obtains at least half of either part from OPT's knapsack. Note that achieving a competitive ratio of 2 matches the lower bound in Theorem 3.1 for general $\hat{\omega}$ (i.e., across all instances).

**Theorem 3.2.** *Given a point prediction,* PP-b *is 2-competitive for* OFKP.

*Proof Sketch.* PP-b selects $\min\{1/2, \hat{\omega}/2\}$ at $\hat{v}$, which is at least half of the profit that OPT obtains from items with value $\hat{v}$. Let $z \leq 1$ be the total weight of items with value $> \hat{v}$. PP-b accepts half of these items above the critical value, again obtaining at least half of OPT's profit. Moreover, we can show that PP-b does not violate the knapsack constraint. The full proof is in Appendix A.6.5. □

We next show how to modify PP-b to achieve the instance-specific lower bound of $1 + \min\{1, \hat{\omega}\}$ from Theorem 3.1.

**PP-a:** *An improved* $1 + \min\{1, \hat{\omega}\}$-*competitive algorithm.* PP-a leverages the observation that it can accept more than half of items with value $> \hat{v}$ when $\hat{\omega}$ is low. However, since $\hat{\omega}$ is unknown, it exploits a "prebuying" strategy, initially selecting entire items with values $> \hat{v}$ before adjusting its selections upon observing items with value $\hat{v}$. This adaptive

---

**Algorithm 1** PP-a: An improved $(1 + \min\{1, \hat{\omega}\})$-competitive algorithm with point prediction

> **Input:** prediction $\hat{v}$
> **Output:** $\{x_i\}_{i \in [n]}$
> **Initialization:** $\hat{\omega}_0 = 0$, $s_0 = 0$
> **for** each item $i$ (with unit value $v_i$ and weight $w_i$) **do**
>    **if** $v_i < \hat{v}$ **then**
>      $\hat{\omega}_i = \hat{\omega}_{i-1}$
>      $x_i = 0$    ▷ item is rejected
>    **else if** $v_i > \hat{v}$ **then**
>      $\hat{\omega}_i = \hat{\omega}_{i-1}$
>      $x_i = w_i/(1 + \hat{\omega}_{i-1})$    ▷ item is partially accepted
>    **else if** $v_i = \hat{v}$ **then**
>      $\hat{\omega}_i = \hat{\omega}_{i-1} + \min\{w_i, 1 - \hat{\omega}_{i-1}\}$
>      $x_i = \frac{\min\{w_i, 1-\hat{\omega}_{i-1}\}}{1+\hat{\omega}_i} - s_{i-1} \cdot \frac{\min\{w_i, 1-\hat{\omega}_{i-1}\}}{1+\hat{\omega}_i}$    ▷
>      item is partially accepted
>    Update $s_i = s_{i-1} + x_i$

---

approach ensures that PP-a selects an appropriate portion on either side of the prediction to achieve the desired bound.

**Theorem 3.3.** *Given a point prediction,* PP-a *is* $(1 + \min\{1, \hat{\omega}\})$-*competitive for* OFKP.

Without prior knowledge of $\hat{\omega}$, PP-a employs a "prebuying" strategy for items with values $> \hat{v}$. The extra capacity allocated to these items always has a higher unit value than $\hat{v}$, allowing PP-a to reduce acceptances from $\hat{v}$ based on how much extra capacity it has allocated in previous items. PP-a maintains a lower bound on $\hat{\omega}$ that increments each time an item with value $\hat{v}$ arrives. This lower bound determines PP-a's strategy for values $> \hat{v}$. The remaining challenge is to ensure that PP-a does not overfill the knapsack, and attains at least $1/(1 + \min\{1, \hat{\omega}\})$ of the profit obtained by OPT. We give the full proof in Appendix A.6.6.

**Intuition behind prebuying.** The key idea of prebuying is that PP-a proactively reserves capacity for items with values strictly greater than the predicted critical value $\hat{v}$, even before observing all such items. This is done to better utilize the knapsack when $\hat{\omega}$, the total weight of items at $\hat{v}$, is small. If the algorithm later encounters many items at $\hat{v}$, it dynamically reduces the amount allocated to higher-value items. This contrasts with reserve-then-greedy strategies, where reservations are fixed in advance.

### 3.3. Trusted Interval Predictions

In this section, we present an algorithm that uses *interval predictions* $[\ell, u]$, where $\hat{v} \in [\ell, u]$ – this is motivated by e.g., uncertainty quantification schemes that provide a bound on the prediction error $\Delta$, such that upper and lower bounds can be modeled as a *confidence interval* $\ell = \tilde{v} - \Delta \leq \hat{v}$ and $u = \tilde{v} + \Delta \geq \hat{v}$, where $\tilde{v}$ is the predicted critical value.

**IPA:** *An interval prediction-based algorithm.* IPA builds

**Algorithm 2** IPA: An Interval-Prediction-Based Algorithm for OFKP

> **Input:** interval prediction $\ell, u$, robust algorithm $\mathcal{A}$ with competitive ratio $\alpha$
> **Output:** online decisions $x_i$s
> Initialize $\mathcal{A}$
> **for** each item $i$ (with unit value $v_i$ and weight $w_i$) **do**
>   **if** $v_i < \ell$ **then**
>     $x_i = 0$     ▷ item is rejected
>   **else if** $v_i > u$ **then**
>     $x_i = \frac{1}{\alpha+1} \times w_i$     ▷ item is partially accepted
>   **else if** $v_i \in [\ell, u]$ **then**
>     Pass item $i$ to algorithm $\mathcal{A}$
>     $x_i = \frac{\alpha}{\alpha+1} \times x_i^{\mathcal{A}}$     ▷ item is partially accepted

on PP−b and is devised to solve OFKP when given interval predictions. It allocates a dedicated portion of the capacity for values higher than $u$, rejects all items with values lower than $\ell$, and employs a sub-algorithm (e.g., ZCL) to solve OFKP within the interval $[\ell, u]$. The results are then combined to yield a competitive result with a competitive ratio of $\alpha + 1$, where $\alpha$ represents the competitive ratio of the sub-algorithm. We summarize pseudocode for IPA in Alg. 2, and give the competitive result for IPA below:

**Theorem 3.4.** *Given an interval prediction $[\ell, u]$ and an algorithm for OFKP with a worst-case competitive ratio of $\alpha$, IPA is $(\alpha + 1)$-competitive for OFKP.*

*Proof Sketch.* For unit values higher than $u$, IPA allocates $1/\alpha+1$ of knapsack capacity. Within the range $[\ell, u]$, it employs a robust sub-algorithm, denoted as $\mathcal{A}$, which is $\alpha$-competitive – it is allocated $\alpha/\alpha+1$ fraction of the knapsack's capacity. Within the interval $[\ell, u]$, obtaining a $\alpha/\alpha+1$ fraction of $\mathcal{A}$'s solution intuitively yields an $(\alpha+1)$-competitive solution against OPT. The remaining challenge is to demonstrate that the bound holds across both parts (i.e., beyond the interval $[\ell, u]$). The full proof is in Appendix A.6.7. □

**Corollary 3.5.** *Given an interval prediction $[\ell, u]$, IPA is $(2 + \ln(u/\ell))$-competitive for OFKP when the sub-algorithm is given by Alg. 4 (ZCL).*

Theorem 3.4 and Corollary 3.5 show that IPA is $(2 + \ln(u/\ell))$-competitive when using an optimal OKP method as the sub-algorithm (e.g., ZCL). A natural question to ask is whether this competitive ratio for trusted interval predictions can be improved upon by any other algorithm – in the following, we answer this in the negative.

**Theorem 3.6.** *Given an interval prediction $[\ell, u]$, no (deterministic or randomized) online algorithm for OFKP can achieve a competitive ratio better than $(2 + \ln(u/\ell))$.*

The result in Theorem 3.6 (full proof in Appendix A.6.2) implies that IPA achieves the optimal competitive ratio for any OKP algorithm using trusted interval predictions.

## 4. Leveraging Untrusted Predictions

In this section, we extend our results to the case of imperfect or *untrusted* predictions. We present MIX, an algorithm that *mixes* the decisions of a prediction-based algorithm with a robust baseline, and prove its consistency and robustness. Furthermore, we give a corresponding lower bound on the best achievable consistency and robustness using a point prediction – this shows that MIX achieves a nearly-optimal consistency-robustness trade-off. Although the technique of combining algorithms is common in the literature (Im et al., 2021; Christianson et al., 2022; Lechowicz et al., 2024), it is not often the case that such a technique approaches the optimal trade-off, making our result particularly noteworthy.

**MIX:** *A robust and consistent algorithm.* MIX combines ZCL, the optimal $(\ln(U/L) + 1)$-competitive OFKP algorithm, with one of the trusted OFKP prediction algorithms presented so far (e.g., PP−a in Alg. 1, IPA in § 3.3). If the prediction is correct, we say that the *"inner prediction algorithm"* ALG is $c$-competitive, where $c$ is the corresponding bound proved in the previous section.

For robustness, we follow prior work (Sun et al., 2021b; Zhou et al., 2008; El-Yaniv et al., 2001) and assume unit values are bounded, i.e., $v_i \in [L, U], \forall i \in [n]$. Note that $L$ and $U$ are not related to the predicted interval $[\ell, u]$. MIX balances between the inner algorithms (worst-case optimized ZCL and prediction-based ALG) by setting a *trust parameter* $\lambda \in [0, 1]$. Both algorithms run in parallel – when an item arrives, MIX receives an item with unit value $v_i$ and weight $w_i$ as input. MIX first "offers" this item to the inner prediction-based ALG, receives its decision $\hat{x}_i$, and then "offers" this item to the inner robust ZCL, receiving a decision $\tilde{x}_i$. Then MIX accepts $x_i = \lambda \hat{x}_i + (1 - \lambda)\tilde{x}_i$ fraction of the item. Note that when $\lambda = 1$, MIX makes the same decisions as the inner prediction ALG, and when $\lambda = 0$, MIX makes the same decisions as the inner robust ZCL. The inner ALG is chosen based on the type of prediction received, e.g., a point or interval prediction. Theorem 4.1 gives bounds on the consistency and robustness of MIX, which gives corresponding consistency-robustness results for the algorithms in the previous section (i.e., for point and interval prediction models, see Theorem 3.3 and Theorem 3.4, respectively).

**Theorem 4.1.** *MIX is $\frac{\ln(U/L)+1}{(1-\lambda)}$-robust and $\frac{c}{\lambda}$-consistent for OFKP for any $\lambda \in (0, 1)$, where $c$ is the competitive ratio of the inner prediction ALG with an accurate prediction.*

*Proof.* MIX's profit is described by $\text{MIX}[\lambda](\mathcal{I}) = \lambda\text{ALG}(\mathcal{I}) + (1 - \lambda)\text{ZCL}(\mathcal{I})$. For consistency, the prediction is correct and ALG is $c$-competitive. Thus, we obtain that $\text{MIX}[\lambda](\mathcal{I}) \geq \lambda\text{ALG}(\mathcal{I})$, and MIX is $\frac{c}{\lambda}$-consistent. For robustness, consider the case where the prediction is wrong. If ALG obtains no profit, we have $\text{MIX}[\lambda](\mathcal{I}) \geq (1 - \lambda)\text{ZCL}(\mathcal{I})$, and MIX is $\frac{\ln(U/L)+1}{(1-\lambda)}$-robust. □

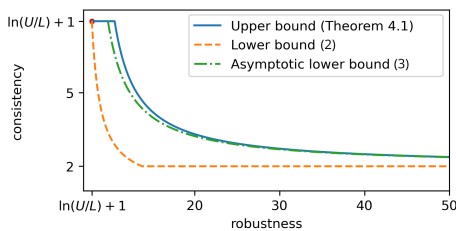

*Figure 1.* The consistency-robustness upper bound of `MIX` using a prediction of the critical value $\hat{v}$ (Theorem 4.1) and the lower bounds ((2) and (3)), with $U/L = 1000$.

We use an example to further clarify the robustness of `MIX`. Set $\lambda = \frac{1}{2}$. Assume that the prediction-based algorithm `ALG` obtains no profit. The `ZCL` algorithm is $c$-competitive. Then, we have

$$\texttt{MIX}[\lambda](\mathcal{I}) = \frac{1}{2} \cdot 0 + \frac{1}{2} \cdot \texttt{ZCL}(\mathcal{I}) = \frac{1}{2} \cdot \frac{\text{OPT}}{c},$$

which shows `MIX` is $2c$-competitive. Thus, for a given non-zero $\lambda$, `MIX` maintains a finite competitive ratio even in worst-case prediction scenarios, highlighting the role of the robust subroutine in guaranteeing performance.

### 4.1. Optimal Consistency-Robustness Trade-Offs

We ask whether any considerable improvement can be made in the consistency-robustness trade-off, i.e., whether an algorithm using succinct predictions can achieve better consistency for a given robustness (or vice versa). In what follows, we show that `MIX` nearly matches the best consistency-robustness trade-off for a critical value prediction $\hat{v}$.

**Theorem 4.2.** *Given an untrusted prediction of the critical value, any deterministic $\gamma$-robust algorithm for `OKP` (where $\gamma \in [\ln(U/L) + 1, \infty)$ is at least $\eta$-consistent for*

$$\eta \geq \max \left\{ 2 - \frac{L}{U}, \ \frac{1}{1 - \frac{1}{\gamma} \ln(U/L)} \right\}. \qquad (2)$$

*Furthermore, in the limit as $U/L \to \infty$, any $2/\lambda$-consistent (for some $\lambda \in (0,1)$) algorithm is at least $\beta$-robust, for*

$$\beta = \frac{1 + \ln(U/L)}{1 - \lambda} - o(1) = \Omega\left(\frac{1 + \ln(U/L)}{1 - \lambda}\right). \qquad (3)$$

This result (full proof in Appendix A.6.3) implies that `MIX` achieves the *nearly optimal trade-off* between consistency and robustness. In Fig. 1, we plot the consistency-robustness upper bound from Theorem 4.1, along with the lower bounds given by (2) and (3). All three bounds behave similarly, exhibiting a convex trade-off. The gap between the upper and lower bound is due to constants in the proof of (3). We disregard consistency $> \ln(U/L) + 1$ since algorithms without predictions (e.g., `ZCL`) give better guarantees.

## 5. Connecting `OIKP` with `OFKP`

In this section, we present a general result that connects the fractional case (`OFKP`) with the integral case (`OIKP`),

---

**Algorithm 3** The `Fr2Int` algorithm

**Input:** `OFKP` algorithm `ALG`, error parameter $\delta$, precision parameter $\epsilon$, bounds $U, L$
**Output:** Online decisions $\{x_i\}_{i \in [n]}$
**Initialization:** Arrays $A[0, \ldots, \lceil \log_{1+\delta}(U/L) \rceil] \leftarrow 0$, $R[0, \ldots, \lceil \log_{1+\delta}(U/L) \rceil] \leftarrow 0$
**for** each item $i$ (with unit value $v_i$ and weight $w_i$) **do**
    Send item $i$ to `ALG` and obtain `OFKP` decision $\tilde{x}_i$
    $j \leftarrow \lceil \log_{1+\delta}(v_i/L) \rceil$
    $R[j] \leftarrow R[j] + \tilde{x}_i \cdot v_i$
    **if** $A[j] < R[j] \cdot \frac{1 - \epsilon(\lceil \log_{1+\delta}(U/L) \rceil + 1)}{1+\delta}$ **then**
        $x_i \leftarrow w_i$   ▷ item is fully accepted
        $A[j] \leftarrow A[j] + x_i \cdot v_i$
    **else**
        $x_i \leftarrow 0$   ▷ item is rejected

---

assuming items have small individual weights and bounded unit values. It is worth mentioning that without any of these two assumptions, there exists no algorithm with meaningful competitive ratio (see Theorem A.1, A.2). In the existing `OKP` literature without predictions, prior work notes that `OIKP` can be solved using discretized variants of the *threshold-based algorithms* for `OFKP` (Zhou et al., 2008; Sun et al., 2021b). However, because the algorithms that we present in § 3 do not use the paradigm of threshold-based design, this straightforward connection is not applicable. Instead, we propose a novel partitioning technique that divides the possible unit values into discrete intervals, allowing a conversion algorithm (`Fr2Int`, see Algorithm 3) to use any `OFKP` algorithm as a subroutine and achieve the same competitive ratio for `OIKP`. We start by formalizing the *value partitioning* component of the algorithm below.

**Definition 5.1** (Value Partitioning). Recall that for `OIKP`, all item unit values lie in the interval $[L, U]$. We divide this interval into $K$ sub-intervals $G_1, G_2, \ldots, G_K$ as follows. Let $\delta > 0$ be small and let $U = L(1+\delta)^K$ for some integer $K$. Then, for any $k \in [K]$, let $G_k = [L(1+\delta)^{k-1}, L(1+\delta)^k)$. Note that by definition, $G_k$ cannot contain values that differ by more than a multiplicative factor of $(1+\delta)$.

We now describe the "conversion" algorithm `Fr2Int` (see Algorithm 3) that connects `OFKP` with `OIKP`. Given any `OFKP` algorithm denoted by `ALG`, `Fr2Int` uses the value partitioning in Definition 5.1 to solve `OIKP` by simulating the fractional actions of `ALG`. When an item arrives for `OIKP` with a value in interval $G_k$, the item is accepted if the sum of values of previously accepted $G_k$ items in the integral knapsack is less than the sum of values from interval $G_k$ in the simulated fractional one. With small item weights, this ensures that the total value in the actual knapsack is close to the simulated one, thus inheriting the competitive bound of `OFKP`. For an `OFKP` algorithm `ALG`, we denote its corresponding `OIKP` variant as `Fr2Int-ALG`.

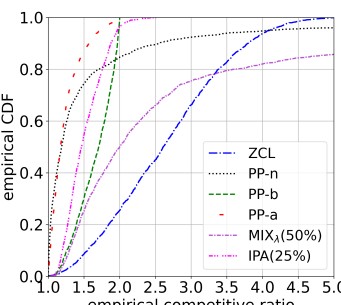 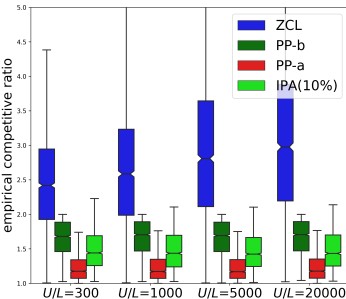 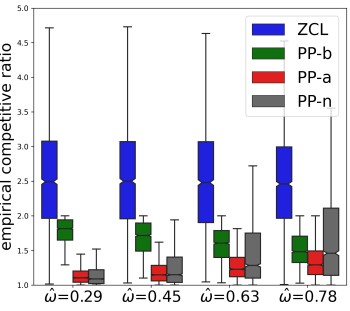

**(a)**            **(b)**            **(c)**

*Figure 2.* **(a)** The CDF plot of the empirical competitive ratio of different algorithms; **(b)** The performance of `PP-b`, `PP-a` and `IPA` versus `ZCL` as $U/L$ varies; and **(c)** The performance of `PP-b`, `PP-a`, and `PP-n` against `ZCL` when $\hat{\omega}$ varies.

**Theorem 5.2.** *Given a $\gamma$-competitive online algorithm* `ALG` *for* `OFKP` *and fixed parameter $\delta > 0$, if the maximum item weight of* `OIKP` *is upper bounded by $\epsilon < 1/\lceil \log_{(1+\delta)} U/L \rceil + 1$, then the algorithm* `Fr2Int-ALG` *is $\gamma \cdot 1+\delta/1-\epsilon(\lceil \log_{(1+\delta)} U/L \rceil + 1)$ competitive for* `OIKP`.

`Fr2Int` extends all our results presented in the context of `OFKP` thus far (e.g., `PP-a` in Theorem 3.3, `IPA` in Theorem 3.4, `MIX` in Theorem 4.1) to the case of `OIKP` with minimal competitive loss. Note that when set $\delta \to 0$ and the maximum item weight $\epsilon$ is small (i.e., $\epsilon < \delta/\lceil \log_{(1+\delta)} U/L \rceil + 1$), `Fr2Int` becomes $\lim_{\delta \to 0} \gamma \cdot 1+\delta/1-\delta$ approaching $\gamma$-competitive. The loss factor $1+\delta/1-\epsilon(\lceil \log_{(1+\delta)} U/L \rceil + 1)$ results from the discretization of the value range and the discrete nature of the item sizes, and it is designed to ensure the feasibility of the integral algorithm after converting from a fractional algorithm.

*Proof Sketch.* `Fr2Int` keeps track of the items accepted in `ALG`'s simulated fractional knapsack with a list $R$ (line 7, Algorithm 3). In lines 8-12, `Fr2Int` checks how the items it has accepted so far (saved in list $A$) compare against the simulated knapsack – this compels `Fr2Int` to accept items that approximate those chosen by `ALG`. The remaining challenge is to show that `Fr2Int` does not overfill the knapsack. For this, the partitioning method allows us to bound the value of a range to its weight. We can show $w(A[j]) \cdot L \cdot (1+\delta)^j \leq A[j] < w(A[j]) \cdot L \cdot (1+\delta)^{j+1}$, which helps us bound the weight using the partitioning idea. The full proof is in Appendix A.6.10. □

We note that by nature of the `OFKP` relaxation, the lower bounds for `OFKP` (namely, Theorem 3.1 and Theorem 4.2) extend to `OIKP`. This follows by observing that on the instances constructed in these proofs, the optimal fractional solution obtains the same value as the optimal integral solution when each fractional item can be subdivided into many small `OIKP` items with the same value density. Since an arbitrary online algorithm can only do worse when it is restricted to accepting entire items, the bounds follow.

## 6. Numerical Experiments

In this section, we present a case study of our proposed algorithms. We compare against baselines that do not use predictions (i.e., `ZCL` (Zhou et al., 2008)), and the existing SEN-TINEL learning-augmented algorithm from prior work (Im et al., 2021), using both synthetic and real datasets.[1] We defer additional experiments and details to Appendix A.7.

**Experimental setup and comparison algorithms.** To validate the performance of our algorithms, we conduct four sets of experiments. In the first set, we use synthetically generated data, where the value and weight of items are randomly drawn from a power-law distribution. Unless otherwise specified, the lowest unit value is $L = 1$, and the highest unit value is $U = 1000$. Weights are drawn from a power law and normalized to the range $[0, 1]$. We report the cumulative density functions (CDFs) of the empirical competitive ratios, which illustrate both the average and worst-case performance of the tested algorithms.

To report the empirical competitive ratio of different algorithms, we solve for the offline optimal solution as described in Appendix A.2. We compare the results of the following algorithms under several experimental settings: **(1)** `ZCL`: the existing optimal algorithm without predictions (Alg. 4); **(2)** `PP-n`: a naïve point-prediction-based algorithm that accepts any item at or above the predicted critical value $\hat{v}$; **(3)** `PP-b`: 2-competitive algorithm for trusted predictions (Alg. 5); **(4)** `PP-a`: $(1 + \min\{1, \hat{\omega}\})$-competitive algorithm (Alg. 1); **(5)** `IPA`: trusted interval-prediction-based algorithm (Alg. 2); **(6)** `MIX`: learning-augmented algorithm (§ 4); **(7)** `Fr2Int-PP-a`: converted `OIKP` version of `PP-a`; and **(8)** SENTINEL algorithm using frequency predictions (Im et al., 2021). For `IPA`, we report the interval prediction range $u - \ell$ as a percentage of $[L, U]$ and set it to 15%, 25%, and 40%. The notation $\text{MIX}_\lambda$ denotes instances of `MIX` under different values of trust parameter $\lambda$, which we set according to $\lambda \in \{0.3, 0.5, 0.9\}$.

---

[1] Our code is available at https://github.com/moreda-a/OKP.

**Experimental results.** Fig. 2(a) reports the cumulative distribution function (CDF) of empirical competitive ratios for six tested algorithms on 2,000 synthetic instances of OFKP. Amongst the prediction-based algorithms, PP-a achieves the best performance in average and worst-case, verifying the results of Theorem 3.3. While PP-n outperforms most others on average (except PP-a), its worst-case performance is worse than ZCL (the optimal algorithm without predictions); this observation verifies bad edge cases for PP-n identified in Theorem A.4. Finally, even with a relatively wide interval prediction, IPA outperforms ZCL in both average and worst-case. We also report the impact of parameters on the performance of the tested algorithms. Given a trusted (i.e., correct) succinct prediction, our theoretical results show that our algorithms (PP-b, PP-a, IPA) can obtain competitive ratios independent of the ratio $U/L$, in contrast to classic online algorithms such as ZCL. In Fig. 2(b), we verify this by varying $U/L$. These results show that the empirical competitive ratio of ZCL increases along with $U/L$, while prediction-based algorithms are robust to these variations. In Fig. 2(c), we vary the value of $\hat{\omega}$ using four values of $0.29, 0.45, 0.63$, and $0.78$. Due to its design (Theorem 3.3), PP-a's performance is better with smaller values of $\hat{\omega}$, while other algorithms see less benefit.

## 7. Conclusion

In this paper, we propose near-optimal learning-augmented algorithms for both fractional and integral knapsack problems using succinct and practical predictions. A number of questions remain for future work. Although the assumption of small item weights is standard in the (integral) OKP literature (Zhou et al., 2008; Sun et al., 2021b; Im et al., 2021), questions remain about an algorithm that uses untrusted predictions without such an assumption. It would be interesting to further explore fundamental trade-offs between consistency and robustness in the case of interval predictions – there is growing literature on *uncertainty quantification* (Sun et al., 2024) in online algorithms that could inform e.g., fine-grained consistency-robustness trade-off bounds based on the error of a given predictor.

## Acknowledgements

This research was supported in part by funding from the National Science Foundation under grant numbers CAREER-2045641, CPS-2136199, and CNS-2325956, and a research gift from Adobe Research. This material is based upon work supported by the U.S. Department of Energy, Office of Science, Office of Advanced Scientific Computing Research, Department of Energy Computational Science Graduate Fellowship under Award Number DE-SC0024386. We thank the anonymous reviewers for their valuable feedback.

## Disclaimers

## Impact Statement

This paper presents work whose goal is to advance the field of Machine Learning. There are many potential societal consequences of our work, none of which we feel must be specifically highlighted here.

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

# A. Appendix

## A.1. Related Work

We make contributions to two lines of work: (i) work on online knapsack, one-way trading, and related problems, e.g., $k$-search, single-leg revenue management, and; (ii) work on online algorithms with advice and learning-augmentation. We describe the relationship to each below.

**Online Knapsack and Online Search.** Our work contributes to the literature on the classic (integral) online knapsack problem first studied in (Marchetti-Spaccamela & Vercellis, 1995), with foundational results given by Zhou et al. (2008). In the past few years, many works have considered variants of this problem, such as removable items (Cygan et al., 2016), item departures (Sun et al., 2022), and generalizations to multidimensional settings (Yang et al., 2021). Closer to this work, several studies have considered the online knapsack problem with additional information or in a learning-augmented setting, including frequency predictions (Im et al., 2021), online learning (Zeynali et al., 2021), advice complexity (Böckenhauer et al., 2014b), and stochastic knapsack (Vondrák & Zenklusen, 2011). Most works in this literature focus on the more difficult case of integral item acceptance; thus, connections between this case and the fractional relaxation in the online setting have been relatively understudied in the literature thus far. A few studies have considered online knapsack with fractional item acceptance under slightly different assumptions, including "online partially fractional knapsack", where items are removable (Noga & Sarbua, 2005), or online fractional knapsack in a random order model (Giliberti & Karrenbauer, 2021), where the arrival order of items is a random permutation (i.e. not adversarial). Sun et al. (2021b) motivate the observation that online fractional knapsack is equivalent to the one-way trading problem with a *rate constraint* for each online price.

The connection between online fractional knapsack (OFKP) and one-way trading (with a rate constraint) motivates a further connection to another track of literature on problems broadly classified as *online search*, including problems such as $1$-max/min search and one-way trading, both first studied by El-Yaniv et al. (2001), $k$-max/min search (Lorenz et al., 2008), and single-leg revenue management (Ball & Queyranne, 2009). In general, OFKP can be understood as a "bridge" between work on online search and online knapsack. Follow-up works have since considered applications of online search problems and additional variants, including cloud pricing (Zhang et al., 2017), electric vehicle charging (Sun et al., 2021b), switching cost of changing decisions (Lechowicz et al., 2023), and learning-augmented versions of both one-way trading (Sun et al., 2021a) and $k$-search (Lee et al., 2022). However, to the best of our knowledge, none of these works consider the impact of rate constraints.

**Learning-Augmented Algorithms.** Learning-augmented algorithm design is an emerging field that incorporates machine-learned predictions about future inputs in algorithm designs, with the goal of matching the good average-case performance of the predictor while maintaining worst-case competitive guarantees. The concepts of consistency and robustness (Lykouris & Vassilvtiskii, 2018; Purohit et al., 2018) give a formal mechanism to quantify the trade-off between following machine-learned predictions and hedging against adversarial inputs, particularly with respect to predictions that are very incorrect. This framework has been applied to a number of online problems, including caching (Lykouris & Vassilvtiskii, 2018), ski-rental (Purohit et al., 2018; Wei & Zhang, 2020; Antoniadis et al., 2021), set cover (Bamas et al., 2020), online selection (Jiang et al., 2021), online matching (Antoniadis et al., 2020b), convex body chasing (Christianson et al., 2022), and metrical task systems (Antoniadis et al., 2020a; Christianson et al., 2023), just to name a few. Most relevant to our setting, it has been explored in the context of online knapsack (Im et al., 2021; Zeynali et al., 2021), unit-profit online knapsack (Boyar et al., 2022), one-way trading (Sun et al., 2021a), and single-leg revenue management (Balseiro et al., 2023).

An overarching goal in this framework is to quantify and match or nearly match an *optimal consistency-robustness trade-off* (Wei & Zhang, 2020). Since gains in consistency generally result in worsened robustness guarantees, it is natural to consider a notion of Pareto-optimality between the two, and ideally to design algorithms that can achieve this best trade-off for any desired robustness or consistency target. Several works have studied optimal trade-offs for different online problems – the closest results to our setting are for one-way trading (Sun et al., 2021a) and single-leg revenue management (Balseiro et al., 2023). However, optimal robustness-consistency trade-offs have not yet been considered in the context of online knapsack – differences in the problem settings of one-way trading and single-leg revenue management result in the substantially different results that we obtain in this work. We also note that the notion of online algorithms with *practical and succinct predictions* has recently been explored in the context of similar problems such as paging (Antoniadis et al., 2023) and non-clairvoyant job scheduling (Benomar & Perchet, 2024).

Learning-augmented algorithms are also closely related to the field of advice complexity, which considers how the performance of an online algorithm can be improved with a specific amount of advice about the input, where the advice is assumed to be correct and is provided by an oracle. This field was first established for paging by (Böckenhauer et al., 2009), with results following for many other online problems (Boyar et al., 2015; Böckenhauer et al., 2014a; Böckenhauer et al., 2011; Komm et al., 2012). Of particular interest to our setting, Böckenhauer et al. (2014b) explore the online knapsack problem with advice, showing that single bit of advice gives a 2-competitive algorithm, but $\Omega(\log n)$ advice bits are necessary to further improve the competitive ratio. They give an online algorithm with competitive ratio $1 + \epsilon$ for any constant $\epsilon > 0$ that uses $O(\log n)$ bits of advice (Böckenhauer et al., 2014b, Theorem 13), where $n$ is the number of items. A similar result was also recently shown for the $k$-search problem by ((Clemente et al., 2022)). To the best of our knowledge, there is no existing work considering the online fractional knapsack problem with advice.

### A.2. Offline Optimal Solution

The offline optimal solution to Equation (1) in the fractional OFKP setting is straightforward to compute. The optimal solution starts by selecting the item with the maximum unit value amongst all $v_i$'s and adds the maximum amount allowed $(w_i)$ to the knapsack. If there is any remaining capacity to fill, the optimal solution then picks the item with the next highest unit value and adds the maximum amount allowed while respecting the capacity constraint. This process is repeated until the knapsack is completely filled.

For ease of analysis, we let $(v_i', w_i')$ denote the values and weights of the items sorted in descending order. That is, $v_1' \geq v_2' \geq \ldots \geq v_n'$. We let $x_i'$ denote the portion of item $(v_i', w_i')$ which is added to the knapsack by some algorithm.

With this notation, the offline optimal solution to (1) can be written as:

$$\mathbf{x}^\star \overset{\text{def}}{=} (x_1^\star, \ldots, x_n^\star) = \left( w_1', w_2', \ldots, w_{r-1}', 1 - \sum_{i=1}^{r-1} w_i', 0, 0, \ldots \right). \tag{4}$$

As seen in (4), the optimal solution selects all the weight of the most valuable items until the knapsack capacity is filled. For lower values, it doesn't acceptance anything. We refer to the last item with a strictly positive acceptance as the $p$th item, which is the maximum value of $j \in [1, n]$ such that $\sum_{i=1}^{j-1} w_i' < 1$.

This optimal solution yields total profit:

$$\text{OPT}(\mathcal{I}) = \sum_{i=1}^{p-1} w_i' v_i' + \left( 1 - \sum_{i=1}^{p-1} w_i' \right) v_p'. \tag{5}$$

Both (4) and (5) hold if the sum of all $w_i$ is greater than or equal to 1, which constitutes the majority of interesting OFKP instances.

In cases where the sum of $w_i$ is less than 1, the optimal solution selects all items, which cannot be described by the above equations. To address this scenario, we introduce an auxiliary item denoted as $v_{n+1}$ with a corresponding weight $w_{n+1}$, where $v_{n+1} = 0$ and $w_{n+1} = 1$. It's important to note that this additional item doesn't affect the profit of any algorithm; rather, it simplifies and maintains consistency in mathematical modeling. In the problematic case, where the sum of $w_i$ is less than 1, $p$ would be equal to $n + 1$, and both (4) and (5) would still remain valid. We note that this case (where the knapsack can accept all items) is somewhat trivial, as the optimal policy simply accepts all items. In the majority of the paper, we implicitly assume that the sum of all $w_i$s is greater than 1.

### A.3. Lower bounds for integral

**Theorem A.1.** *There is no deterministic algorithm for the online integral knapsack problem (OIKP) that has a meaningful competitive ratio with or without a critical value prediction when only the bounded unit value assumption holds.*

*Proof.* Consider the following two input instances, each with a critical value $\hat{v} = 1$ and a total critical weight of $2\kappa$ at the critical value:

$$\mathcal{I}_1: \quad \text{number of items } n = 1, \quad (v_1, w_1) = (1, 2\kappa).$$
$$\mathcal{I}_2: \quad \text{number of items } n = 2, \quad (v_1, w_1) = (1, 2\kappa), \quad (v_2, w_2) = (U, 1 - \kappa).$$

where $\kappa \to 0$. The integral offline optimal solutions for the two instances are $\text{OPT}(\mathcal{I}_1) = 2\kappa$ and $\text{OPT}(\mathcal{I}_2) = (1-\kappa)U$, respectively.

Since both instances have the same critical value and are thus given the same prediction, and since they share the same first item, any online algorithm (whether it uses a prediction or not) will make the same decision regarding the first item, regardless of which instance is presented.

Since this problem is integral, the algorithm must either accept or reject the first item. If it accepts the first item, there is not enough capacity for the second item. In this case, the competitive ratio for $\mathcal{I}_2$ is $c(\mathcal{I}_2) = \frac{U(1-\kappa)}{2\kappa}$, which tends to infinity as $\kappa$ approaches $0$.

In the alternative case, if the algorithm rejects the first item, then in $\mathcal{I}_1$, the value obtained is $0$, meaning that the competitive ratio for $\mathcal{I}_1$ is also undefined. Thus, in both cases, the competitive ratio of the algorithm is not meaningful. $\qquad\square$

**Theorem A.2.** *There is no deterministic algorithm for the online integral knapsack problem (*OIKP*) that has a meaningful competitive ratio with or without a critical value prediction when only the small weights assumption holds.*

*Proof.* Consider the following set of input instances, each with a critical value $\hat{v}$ and a total weight of $1$ at the critical value, where $\kappa > 0$ is the (small) weight of each item:

$$\mathcal{I}_1 : \quad \text{number of items } n = 1/\kappa, \text{ each item value and weight:} \quad (v_i, w_i) = (\hat{v}, \kappa).$$

For this first instance, any $c$-competitive algorithm ALG must accept at least $1/c$ of the items. Other instances have the first instance as the prefix, with the second part defined as follows:

$$\forall i \in [1, 1/\kappa - 1], \ \mathcal{I}_i : \text{ total no. of items } n = 1/\kappa + i - 1, \ (v_{j+1/\kappa}, w_{j+1/\kappa}) = \left( \frac{c(c+1)^{i-2}}{\kappa} \hat{v}, \kappa \right) : i > 1/\kappa.$$

Since the critical value is the same across all instances $\{\mathcal{I}_i\}$, and instance $\mathcal{I}_i$ is a prefix of all instances $\mathcal{I}_j : j > i$, we can say that any decision made by a deterministic algorithm on the shorter instance must be repeated in the larger instance as the early part of the input arrives.

Note that the total value of all items in each instance $i$ is $(c+1)^{i-1}\hat{v}$. Ignoring the knapsack's capacity limit, ALG can at most accept this amount. However, in instance $i+1$, the last item's actual value is $c(c+1)^{i-1} \cdot \frac{1}{\kappa} \times \kappa = c(c+1)^{i-1}$, which is $c$ times higher than the sum of all previous values. This implies that, in order for the algorithm to be competitive, it must pick the last item of instance $i+1$. The same argument can be made for all $i$, meaning that the algorithm must accept all items; otherwise, ALG will not be $c$-competitive for at least one instance $\mathcal{I}_i$.

However, if ALG was to accept all of the items that are necessary for $c$-competitiveness on all instances $\{\mathcal{I}_i\}$ simultaneously, it would require a capacity of $1/c + (1/\kappa - 1)\kappa$, which simplifies to $1/c + 1 - \kappa$. As $\kappa \to 0$, this is greater than $1$. Therefore, there is no fixed $c$ such that an arbitrary ALG can be $c$-competitive. $\qquad\square$

## A.4. Deferred Pseudocode for Existing Optimal Online Algorithm (ZCL)

**Pseudocode for ZCL.** Here we give the pseudocode for the baseline algorithm ZCL– this is the known optimal deterministic algorithm for OKP (i.e., both the fractional (OFKP) and integral (OIKP) cases) without predictions. ZCL takes a threshold function $\phi(z) : [0, 1] \to [L, U]$ as input (note that the maximum and minimum unit values $U$ and $L$ are assumed to be known). $\phi(z)$ is understood as the pseudo price of packing a small amount of item when the knapsack's current *utilization* (i.e. the fraction of total capacity filled with previously accepted items) is $z$. The algorithm determines the decision $x_i$ by solving an optimization problem $x_i = \arg\max_{x_i \in \mathcal{X}_i \cap [0, 1-z]} x_i v_i - \int_z^{z+x_i} \phi(u) du$, where recall that $\mathcal{X}_i$ is the feasible decision set defined in § 2. For OIKP, the algorithm will admit the item if $v_i \geq \phi(z)$ and there is sufficient remaining capacity $w_i \leq 1 - z$; for OFKP, the algorithm will continuously admit the item until one of the following occurs: (i) the utilization reaches $\phi^{-1}(v_i)$; (ii) the entire item is admitted; or (iii) the knapsack capacity is used up. Notice the threshold function $\phi$ is the only design space for Algorithm 4. (Sun et al., 2021b) shows that the optimal competitive ratio can be attained when $\phi$ is carefully designed as follows.

**Lemma A.3** (Theorem 3.5 in (Sun et al., 2021b), Theorem 2.1 in (Zhou et al., 2008)). *When the unit value of items is bounded within $[L, U]$, Algorithm 4 is $(1 + \ln(U/L))$-competitive for OFKP and OIKP (integral with small item weights)*

---

**Algorithm 4** `ZCL`: An Online Threshold-Based Algorithm for `OKP` Without Prediction

---

**Input:** threshold function $\phi(z)$
**Output:** online decisions $\{x_i\}_{i \in [n]}$
**Initialization:** knapsack utilization $z^{(0)} \leftarrow 0$
**for** each item $i$ (with unit value $v_i$ and weight $w_i$) **do**
$\quad \mathcal{X}_i \leftarrow \{0, w_i\}$ or $[0, w_i]$ {feasible set: discrete for `OIKP`, continuous for `OFKP` }
$\quad$ **if** $v_i < \phi(z^{(i-1)})$ **then**
$\quad\quad x_i \leftarrow 0$
$\quad$ **else**
$\quad\quad x_i \leftarrow \arg\max_{x_i \in \mathcal{X}_i \cap [0, 1-z^{(i-1)}]} \left( x_i v_i - \int_{z^{(i-1)}}^{z^{(i-1)}+x_i} \phi(u)\, du \right)$
$\quad$ Update $z^{(i)} \leftarrow z^{(i-1)} + x_i$

---

---

**Algorithm 5** `PP-a`: A Basic 2-Competitive Algorithm for `OFKP` with Trusted Prediction

---

**Input:** prediction $\hat{v}$
**Output:** online decisions $\{x_i\}$
**for** each item $i$ (with unit value $v_i$ and weight $w_i$) **do**
$\quad$ **if** $v_i < \hat{v}$ **then**
$\quad\quad x_i \leftarrow 0$ {item is rejected}
$\quad$ **else if** $v_i > \hat{v}$ **then**
$\quad\quad x_i \leftarrow \frac{w_i}{2}$ {item is partially accepted}
$\quad$ **else if** $v_i = \hat{v}$ **then**
$\quad\quad$ temp $\leftarrow \min(w_i, 1 - \hat{\omega})$
$\quad\quad \hat{\omega} \leftarrow \hat{\omega} + $ temp
$\quad\quad x_i \leftarrow \frac{\text{temp}}{2}$ {item is partially accepted}

---

*when the threshold is given by*

$$\phi(z) = \begin{cases} L & z \in \left[0, \frac{1}{1+\ln(U/L)}\right) \\ L \exp\left((1 + \ln(U/L))z - 1\right) & z \in \left[\frac{1}{1+\ln(U/L)}, 1\right] \end{cases}. \tag{7}$$

*Further, no online algorithm can achieve a competitive ratio smaller than $1 + \ln(U/L)$.*

### A.5. Deferred Pseudocode for § 3 (Trusted Predictions)

**`PP-n`:** *A naïve trusted-prediction algorithm:* We consider a naïve "greedy" algorithm that takes a prediction on $\hat{v}$ as input. The first algorithm rejects any items with unit value $< \hat{v}$ and fully accepts any item with unit value $\geq \hat{v}$ until the capacity limit. In Theorem A.4, we show that `PP-n` fails to achieve a meaningful improvement in the worst-case competitive ratio (i.e., consistency since we assume the prediction is correct). We prove the following result in § A.6.4. The second way of defining is to reject items with value $\leq \hat{v}$ and accept items with value $> \hat{v}$. This algorithm is not competitive too since there could be no item with greater value than $\hat{v}$ which means you won't accept anything.

**Theorem A.4.** `PP-n` *that fully trusts the prediction is $^U/_L$-competitive in the worst case.*

It is worth mentioning that this algorithm's experimental results show the benefits of using critical value prediction even without improvements, as seen in Figure A6. However, it falls short of providing meaningful theoretical results in special cases.

**Pseudocode for `PP-a`.** Here we give the pseudocode for the `PP-a` algorithm (Algorithm 5) discussed in § 3.2 and Theorem 3.2.

**A.6. Proofs**

A.6.1. PROOF OF THEOREM 3.1

*Proof.* Consider the following two input instances, each with critical value $\hat{v} = 1$ and total weight $\hat{\omega}$ on the critical value:

$$\mathcal{I}_1: \quad n = 1, \quad (v_1, w_1) = (1, \hat{\omega}),$$
$$\mathcal{I}_2: \quad n = 2, \quad (v_1, w_1) = (1, \hat{\omega}), \quad (v_2, w_2) = (U, 1 - \epsilon).$$

where $U \to +\infty$ and $\epsilon \to 0$. The offline optimal solutions for the two instances are $\text{OPT}(\mathcal{I}_1) = \min\{1, \hat{\omega}\}$ and $\text{OPT}(\mathcal{I}_2) = (1 - \epsilon)U + \epsilon$, respectively.

Since both instances have the same critical value and thus are given the same prediction, and since they have the same first item, any online algorithm will make the same decision for the first item regardless of which instance is presented. Let $X \in [0, \min\{1, \hat{\omega}\}]$ denote the decision for the first item from a randomized algorithm, where $X$ is a random variable with probability density function $f(x)$. The expected returns of the randomized algorithm over the two instances can be derived as

$$\mathbb{E}[\text{ALG}(\mathcal{I}_1)] = \int_0^{\min\{1,\hat{\omega}\}} f(x)x dx = \mathbb{E}[X],$$

$$\mathbb{E}[\text{ALG}(\mathcal{I}_2)] \leq \int_0^{\min\{1,\hat{\omega}\}} f(x)[x + (1-x)U] dx = \mathbb{E}[X] + (1 - \mathbb{E}[X])U.$$

where $\mathbb{E}[X] \in [0, \min\{1, \hat{\omega}\}]$. As $U \to +\infty$ and $\epsilon \to 0$, the competitive ratio of the algorithm is thus

$$\max\left\{\frac{\text{OPT}(\mathcal{I}_1)}{\mathbb{E}[\text{ALG}(\mathcal{I}_1)]}, \frac{\text{OPT}(\mathcal{I}_2)}{\mathbb{E}[\text{ALG}(\mathcal{I}_2)]}\right\} \geq \max\left\{\frac{\min\{1,\hat{\omega}\}}{\mathbb{E}[X]}, \frac{1}{1 - \mathbb{E}[X]}\right\} \geq 1 + \min\{1, \hat{\omega}\}.$$

where the final bound follows by observing that the competitive ratio is minimized when $\frac{\min\{1,\hat{\omega}\}}{\mathbb{E}[X]} = \frac{1}{1-\mathbb{E}[X]}$, which occurs when $\mathbb{E}[X] = \frac{\min\{1,\hat{\omega}\}}{1+\min\{1,\hat{\omega}\}}$ Thus, the competitive ratio of any online randomized algorithm is at least $1 + \min\{1, \hat{\omega}\}$. This completes the proof. $\square$

A.6.2. PROOF OF THEOREM 3.6

*Proof.* Based on the matching lower bound from the ZCL paper (Zhou et al., 2008), for any (possibly randomized) algorithm with known lower and upper bounds $\ell$ and $u$ on value densities, there exists an input sequence, denoted by $\mathcal{I}$, such that the expected total value of the algorithm is at most $\frac{Z}{1+\ln(u/\ell)}$, where $Z = \text{OPT}(\mathcal{I})$ is the offline optimal value under the instance $\mathcal{I}$. Note that in this instance, the last item is with the maximum value density within $[\ell, u]$ and with weight 1.

Let $\mathcal{J}$ denote a new instance by appending one item with value $U$ and weight $1 - \varepsilon$ (where $\varepsilon > 0$ and $\varepsilon \to 0$) to the instance $\mathcal{I}$. Note that the instance $\mathcal{J}$ has the same prediction interval as the instance $\mathcal{I}$. The offline optimal value $\text{OPT}(\mathcal{J}) \geq (1 - \varepsilon)U \to U$. For an online algorithm $\mathcal{A}$ given an interval prediction $[\ell, u]$, let $M \in [0, 1]$ denote total amount of admitted items under the instance $\mathcal{I}$ and let $f(m)$ denote the probability distribution of the random variable $M$. The total value obtained by the online algorithm can be calculated as follows.

$$\mathbb{E}[\mathcal{A}(\mathcal{I})] \leq \int_0^1 f(m)m \cdot \left(\frac{Z}{1+\ln(u/\ell)}\right) dm,$$

$$= \mathbb{E}[M] \cdot \frac{Z}{1+\ln(u/\ell)}.$$

$$\mathbb{E}[\mathcal{A}(\mathcal{J})] \leq \int_0^1 f(m)\left[m \cdot \frac{Z}{1+\ln(u/\ell)} + (1-m)U\right] dm,$$

$$= \mathbb{E}[M] \cdot \frac{Z}{1+\ln(u/\ell)} + (1 - \mathbb{E}[M])U.$$

where $\mathbb{E}[M] \in [0, 1]$. As $U \to \infty$, the competitive ratio of the algorithm is lower bounded by

$$\max\left\{\frac{\text{OPT}(\mathcal{I})}{\mathbb{E}[\mathcal{A}(\mathcal{I})]}, \frac{\text{OPT}(\mathcal{J})}{\mathbb{E}[\mathcal{A}(\mathcal{J})]}\right\} \geq \max\left\{\frac{Z}{\mathbb{E}[M] \cdot \frac{Z}{1+\ln(u/\ell)}}, \quad \frac{1}{1 - \mathbb{E}[M]}\right\} \geq 2 + \ln(u/\ell).$$

$\square$

### A.6.3. PROOF OF THEOREM 4.2

We divide the proof into the proofs of the following two lemmas, provide the detailed proof of the asymptotic lower bound in Lemma A.6, and postpone the proof of the lower bound in Lemma A.5 to § A.6.8.

**Lemma A.5.** *Given an untrusted prediction of the critical value, any deterministic $\gamma$-robust learning-augmented algorithm for OKP (where $\gamma \in [\ln(U/L) + 1, \infty)$ is at least $\eta$-consistent, where $\eta$ is defined in (2).*

**Lemma A.6.** *In an asymptotic regime for OKP where $U/L \to \infty$, any deterministic algorithm given an untrusted prediction of the critical value that is $2x$-consistent ($x \in (1, \infty)$) is at least $\Omega\left(\frac{1+\ln(U/L)}{1-\frac{1}{x}}\right)$-robust.*

*Proof of Lemma A.6.* Suppose an algorithm for OKP is $\beta$-consistent and $\alpha$-robust, where $\beta = 2x$ (noting that 2-consistency is a firm lower bound by Lemma A.5). Consider the following instance:

- The first batch of items to arrive have value $L$. There are $1/\epsilon$ of these, each with weight $\epsilon$.

- The second batch of items has value $AL$, where $A \geq 1$. There are $1/\epsilon - 1$ many of these items, each with weight $\epsilon$.

- The third batch of items has value $BL$, where $B > A \geq 1$. There are many of these items, and each has weight epsilon.

First, we consider **consistency** on the first two batches of items, letting $\hat{v} = L$. Note that $\text{OPT} \to AL$ as $\epsilon \to 0$, and observe that ALG must satisfy the following:

$$\text{ALG(first two batches)} = \frac{1}{2x}L + \left(\frac{1}{2x} - \frac{1}{2Ax}\right)AL.$$

It follows that ALG must use $\left(\frac{1}{x} - \frac{1}{2Ax}\right)$ of its knapsack capacity during these first two batches of items in order to stay $2x$-consistent. Also note when $AL \gg L$, $\left(\frac{1}{x} - \frac{1}{2Ax}\right) \to 1/x$.

Next, we consider **robustness** on all three batches of items. As previously, ALG uses $\left(\frac{1}{x} - \frac{1}{2Ax}\right)$ of its knapsack capacity during the first two batches of items, leaving $\left(1 - \frac{1}{x} + \frac{1}{2Ax}\right)$ capacity remaining for robustness. We will assume that $BL \gg AL$, implying that the optimal strategy is to run an existing optimal OKP algorithm (i.e., ZCL) with the remaining capacity. Let $\ln(U/BL) + 1$ be the competitive ratio of this "inner" ZCL algorithm, and note that the algorithm must accept (within the available capacity) a $\frac{1}{1+\ln(U/BL)}$ fraction of the items with value $BL$. This gives us the following for ALG on all three batches:

$$\text{ALG(all three batches)} = \frac{1}{2x}L + \left(\frac{1}{2x} - \frac{1}{2Ax}\right)AL + \left(1 - \frac{1}{x} + \frac{1}{2Ax}\right) \cdot \left(\frac{BL}{1 + \ln(U/BL)}\right).$$

Note that $\texttt{OPT} \to BL$ on the sequence that includes the third batch. Then we have the following for the robustness ratio $\beta$:

$$
\begin{aligned}
\beta &= \frac{BL}{\frac{1}{2x}L + \left(\frac{1}{2x} - \frac{1}{2Ax}\right)AL + \left(1 - \frac{1}{x} + \frac{1}{2Ax}\right) \cdot \left(\frac{BL}{1+\ln(U/BL)}\right)}, \\
&= \frac{1 + \ln(U/L)}{1 - 1/x}, \\
&\times \left(1 - \frac{A(\ln(U/L)+1) + 2B - 2Bx + 2x\left(1 - \frac{1}{x} + \frac{1}{2Ax}\right)\left(\frac{B(\ln(U/L)+1)}{1+\ln(U/BL)}\right)}{A(\ln(U/L)+1) + 2x\left(1 - \frac{1}{x} + \frac{1}{2Ax}\right)\left(\frac{B(\ln(U/L)+1)}{1+\ln(U/BL)}\right)}\right), \\
&= \frac{1 + \ln(U/L)}{1 - 1/x}, \\
&- \left(1 - \frac{2Bx - 2B}{A(\ln(U/L)+1) + \left(2Bx - 2B + \frac{B}{A}\right) \cdot \left(\frac{\ln(U/L)+1}{1+\ln(U/BL)}\right)}\right).
\end{aligned}
$$

Let $x$ be a constant (i.e., independent of $A$ and $B$), let $F(A, B, U, L, x) = 1 - \frac{2Bx-2B}{A(\ln(U/L)+1)+\left(2Bx-2B+\frac{B}{A}\right)\left(\frac{\ln(U/L)+1}{1+\ln(U/BL)}\right)}$, and consider the limit as $A \to \infty$, $B \to \infty$, $B/A \to \infty$, and $U/L \to \infty$. Noting that $\ln(U/BL) \approx \ln(U/L)$ as $U \to \infty$, we have that $F(A, B, U, L, x) \to o(1)$ under the above conditions. Thus, we have the following, completing the lemma:

$$
\beta \to \frac{1 + \ln(U/L)}{1 - 1/x} - o(1) \to \Omega\left(\frac{1 + \ln(U/L)}{1 - 1/x}\right) = \Omega\left(\frac{1 + \ln(U/L)}{1 - \lambda}\right).
$$

where $\lambda = 1/x \in (0, 1)$. $\qquad \square$

The statement of Theorem 4.2 follows by combining the results of Lemma A.5 and Lemma A.6.

### A.6.4. PROOF OF THEOREM A.4

*Proof.* Denote the prediction received by the algorithm as $\hat{v}$, for any valid $\texttt{OFKP}$ instance $\mathcal{I}$.

Consider the following special instance in $\Omega$.

$$
\mathcal{I}: \quad n = 2, \quad (v_1, w_1) = (L, 1), \quad (v_2, w_2) = (U, 1 - \epsilon). \tag{10a}
$$

where $U \to +\infty$ and $\epsilon \to 0$. Note that the offline optimal return of this instance is $\texttt{OPT}(\mathcal{I}) = U(1 - \epsilon) + L(\epsilon)$, and $\hat{v} = L$.

Observe that the naïve algorithm will receive the exact value of $\hat{v}$ and greedily accept any items with unit value at or above $\hat{v}$. Then the first item with $(v_1, w_1) = (L, 1)$ will fill the online algorithm's knapsack, and the competitive ratio can be derived as

$$
\frac{\texttt{OPT}(\mathcal{I})}{\texttt{ALG}(\mathcal{I})} = \frac{U(1 - \epsilon) + L(\epsilon)}{\hat{v}} = \frac{U(1 - \epsilon) + L(\epsilon)}{L}.
$$

As $\epsilon \to 0$, the right-hand side implies that the competitive ratio is bounded by $U/L$. Since an accurate prediction has $\hat{v} \in [L, U]$ by definition, this special instance also gives the worst-case competitive ratio over all instances. $\qquad \square$

### A.6.5. PROOF OF THEOREM 3.2

*Proof.* Before starting the proof, we define a new notation for the optimal offline solution. Let's assume that there are $q - 1$ items with strictly greater values $v_i' > \hat{v}$ and items $q$ to $p - 1$ are items with unit value $\hat{v}$ ($q$ can equal $p - 1$, implying there are zero such items) . We can rewrite (5) as follows:

$$
\texttt{OPT}(\mathcal{I}) = \sum_{i=1}^{q-1} w_i'v_i' + \sum_{i=q}^{p-1} w_i'\hat{v} + \left(1 - \sum_{i=1}^{p-1} w_i'\right)\hat{v}. \tag{11}
$$

using the above notation we define $\hat{\omega}$ as:

$$\hat{\omega} := \sum_{i=q}^{r} w'_i. \tag{12}$$

where $r$ is largest number which $v'_i = \hat{v}$ and is greater or equal to $p$. Recall that $\hat{v} := v'_p$.

With this notation in place, we can proceed with the proof.

As described in Algorithm 5, each item $i$ with a value less than the prediction $\hat{v}$ is ignored. If the value is greater than the prediction, half of its weight is selected. If the prediction is equal to the value, we will select half of it and ensure that the sum of all selections with a value equal to the prediction doesn't exceed $1/2$. We first show that Algorithm 5 outputs a feasible solution, i.e., that $\sum_{i=1}^{n} x_i \leq 1$. We derive the following equation:

$$\sum_{i=1}^{n} x_i = \left( \sum_{v_i < \hat{v}} x_i \right) + \left( \sum_{v_i = \hat{v}} x_i \right) + \left( \sum_{\hat{v} < v_i} x_i \right). \tag{13}$$

The first sum is equal to zero, since the algorithm doesn't select any items with $v_i < \hat{v}$. The second sum considers $v_i = \hat{v}$, and the algorithm selects half of every weight $w_i$ unless $\hat{\omega}$ is greater than $1/2$. The algorithm ensures it doesn't select more than $1/2$ by definition in line 9, which checks whether to take half of an item and not exceed the remaining amount from $1/2$. For $v'_i \geq v'_p$, Algorithm 5 sets $x_i$ to $w_i/2$.

$$\sum_{i=1}^{n} x_i = 0 + \min\left( \frac{\hat{\omega}}{2}, \frac{1}{2} \right) + \left( \sum_{\hat{v} < v_i} \frac{w_i}{2} \right). \tag{14}$$

The last sum is $1/2$ times the of the sum of $w_i$s for any items with a value greater than $\hat{v}$. If we look at (11), all of these $w_i$ are completely selected in the optimal offline solution (in the first part of the equation). Thus, their sum is less than or equal to 1, since the optimal solution is feasible: $\sum_{\hat{v} < v_i} w_i = \sum_{i=1}^{q-1} w'_i \leq 1$.

$$\sum_{i=1}^{n} x_i \leq \frac{1}{2} + \frac{1}{2} \cdot \left( \sum_{\hat{v} < v_i} w_i \right) \leq \frac{1}{2} + \frac{1}{2} = 1. \tag{15}$$

Equation 15 proves that the solution from Algorithm 5 is feasible. We next calculate the profit obtained by Algorithm 5 and bound its competitive ratio. The profit can be calculated using (1) as:

$$\text{ALG}(\mathcal{I}) = \min\left( \frac{\hat{\omega}}{2}, \frac{1}{2} \right) \cdot \hat{v} + \left( \sum_{\hat{v} < v_i} \frac{w_i}{2} \cdot v_i \right). \tag{16}$$

Looking at (11), we claim that the second part of (16) is $\frac{1}{2} \cdot \sum_{i=1}^{q-1} w'_i v'_i$.

To calculate the competitive ratio, we give a bound on $\text{OPT}(\mathcal{I})/\text{ALG}(\mathcal{I})$ by substituting (11) and (16) into the definition of CR (i.e. $\text{OPT}/\text{ALG}$), obtaining the following:

$$\text{CR} = \max_{\mathcal{I} \in \Omega} \frac{\sum_{i=1}^{q-1} w'_i v'_i + \sum_{i=q}^{p-1} w'_i \hat{v} + \left( 1 - \sum_{i=1}^{p-1} w'_i \right) \hat{v}}{\sum_{i=1}^{q-1} \frac{w'_i}{2} v'_i + \min\left( \frac{\hat{\omega}}{2}, \frac{1}{2} \right) \cdot \hat{v}}. \tag{17}$$

$$\text{CR} = \max_{\mathcal{I} \in \Omega} \left( 2 \cdot \frac{\sum_{i=1}^{q-1} w'_i v'_i + \sum_{i=q}^{p-1} w'_i \hat{v} + \left( 1 - \sum_{i=1}^{p-1} w'_i \right) \hat{v}}{\sum_{i=1}^{q-1} w'_i v'_i + \min\left( \hat{\omega}, 1 \right) \cdot \hat{v}} \right). \tag{18}$$

Here we prove that the numerator is less than or equal to the denominator, which will give us (19).

$$\text{CR} = \max_{\mathcal{I} \in \Omega} \left( 2 \cdot \frac{\sum_{i=1}^{q-1} w'_i v'_i + \sum_{i=q}^{p-1} w'_i \hat{v} + \left( 1 - \sum_{i=1}^{p-1} w'_i \right) \hat{v}}{\sum_{i=1}^{q-1} w'_i v'_i + \min\left( \hat{\omega}, 1 \right) \cdot \hat{v}} \right) \leq 2. \tag{19}$$

We argue two cases: first, if $\hat{\omega} < 1$, then Algorithm 5 will always select half of every item with value $\hat{v}$. Then, by rewriting the value of $\hat{\omega}$ we have:

$$\sum_{i=1}^{q-1} w_i' v_i' + \min(\hat{\omega}, 1) \cdot \hat{v} = \sum_{i=1}^{q-1} w_i' v_i' + \left(\sum_{i=q}^{u} w_i'\right) \hat{v} = \sum_{i=1}^{q-1} w_i' v_i' + \sum_{i=q}^{p-1} w_i' \hat{v} + w_p' \hat{v} + \sum_{i=p+1}^{u} w_i' \hat{v}. \tag{20}$$

Using the fact that the definition of $x_p^\star$ in (4) and LP constraint (1), we claim $1 - \sum_{i=1}^{p-1} w_i' = x_p^* \leq w_p$. Thus, (20) is greater than the numerator in (19).

In the second case, if $\hat{\omega} \geq 1$, then the algorithm will stop selecting items with a value of $\hat{v}$ after it reaches a capacity of $1/2$ for those items. The optimal offline solution is feasible, so $\sum_{i=1}^{p-1} w_i' + 1 - (\sum_{i=1}^{p-1} w_i') \leq 1$, which implies that the following is also true: $\sum_{i=q}^{p-1} w_i' + 1 - (\sum_{i=1}^{p-1} w_i') \leq 1$. Using this, we can rewrite the numerator:

$$\sum_{i=1}^{q-1} w_i' v_i' + \sum_{i=q}^{p-1} w_i' \hat{v} + \left(1 - \sum_{i=1}^{p-1} w_i'\right) \hat{v} \leq \sum_{i=1}^{q-1} w_i' v_i' + 1 \cdot \hat{v}. \tag{21}$$

Which subsequently implies that the numerator is less than or equal to the denominator.

Both cases have been proven, completing the proof of (19) – $\mathrm{PP-a}$ is 2-competitive. □

### A.6.6. PROOF OF THEOREM 3.3

Let $\mathcal{I} := \{(w_i, v_i)\}_{i \in [n]}$ denote an instance for $\mathrm{OFKP}$, where the value density $v_i \geq \hat{v}, \forall i \in [n]$. It is without loss of generality to focus on $\mathcal{I}$ since both offline algorithm and $\mathrm{PP-a}$ ignore items with value density smaller than $\hat{v}$. To distinguish the items with critical value density $\hat{v}$ and those with value density greater than $\hat{v}$. Define $\mathcal{N}_i^c := \{j \in [i] : v_j = \hat{v}\}$ and $\mathcal{N}_i^o := \{j \in [i] : v_j > \hat{v}\}$ as the sets of items (up to the $i$-th item) whose value densities are equal to and greater than $\hat{v}$, respectively. Then the offline optimal value under instance $\mathcal{I}$ can be shown as

$$\mathrm{OPT}(\mathcal{I}) = \sum_{i \in \mathcal{N}_n^o} v_i w_i + \hat{v}(1 - \sum_{i \in \mathcal{N}_n^o} w_i). \tag{22}$$

where the first part is total value of items with value densities greater than $\hat{v}$, and the second part is the value of items with density $\hat{v}$, filling the remaining knapsack capacity.

Let $x_i$ denote the online admission solution of item $i$. We first show that the online solution by $\mathrm{PP-a}$ is feasible.

**Lemma A.7.** *The online solution of* $\mathrm{PP-a}$ *is feasible for* $\mathrm{OFKP}$.

*Proof of Lemma A.7.* We use $s_i = \sum_{j \in [i]} x_j$ to denote the cumulative amount of admitted items by $\mathrm{PP-a}$, up to item $i$. The goal is to prove $s_n \leq 1$. We claim that the cumulative amount of admitted items by $\mathrm{PP-a}$ is

$$s_i = \frac{\hat{\omega}_i + \tilde{\omega}_i}{1 + \hat{\omega}_i}, \forall i \in [n]. \tag{23}$$

where $\hat{\omega}_i = \min\{\sum_{j \in \mathcal{N}_i^c} w_j, 1\}$ denotes the cumulative amount of item weights with critical value density $\hat{v}$, upper bounded by 1, and $\tilde{\omega}_i = \sum_{j \in \mathcal{N}_i^o} w_j$ denotes the cumulative amount of item weights with critical value density greater than $\hat{v}$. Note that $\tilde{\omega}_n \leq 1$ by definition of the critical value. Thus, if (23) holds, the online solution of $\mathrm{PP-a}$ is feasible since $s_n = \frac{\hat{\omega}_n + \tilde{\omega}_n}{1 + \hat{\omega}_n} \leq 1$.

In the following, we prove (23) by induction.

*Base Case:* $i = 1$. Initially, $\hat{\omega}_0 = 0$ and $s_0 = 0$. If item 1 is not with critical value, i.e., $1 \in \mathcal{N}_n^o$, then by $\mathrm{PP-a}$, we have $\hat{\omega}_1 = \hat{\omega}_0 = 0$ and $s_1 = x_1 = w_1 = \tilde{\omega}_1$, which satisfies (23). If item 1 is with critical value, i.e., $1 \in \mathcal{N}_n^c$, then $\hat{\omega}_1 = w_1$, $\tilde{\omega}_1 = 0$, and $s_1 = x_1 = \frac{\hat{\omega}_1}{1 + \hat{\omega}_1}$, which satisfies (23).

*Induction Step:* $i \geq 2$. Suppose $s_{i-1} = \frac{\hat{\omega}_{i-1} + \tilde{\omega}_{i-1}}{1 + \hat{\omega}_{i-1}}$, we next show that $s_i = \frac{\hat{\omega}_i + \tilde{\omega}_i}{1 + \hat{\omega}_i}$. Consider the following two cases.

*Case (i):* If item $i \in \mathcal{N}_n^o$, we have $\hat{\omega}_i = \hat{\omega}_{i-1}$ and $x_i = \frac{w_i}{1+\hat{\omega}_{i-1}}$ by PP-a. This gives

$$s_i = s_{i-1} + x_i = \frac{\hat{\omega}_{i-1} + \tilde{\omega}_{i-1}}{1 + \hat{\omega}_{i-1}} + \frac{w_i}{1 + \hat{\omega}_{i-1}} = \frac{\hat{\omega}_i + \tilde{\omega}_i}{1 + \hat{\omega}_i}. \tag{24}$$

*Case (ii):* If item $i \in \mathcal{N}_n^c$, then we have $\tilde{\omega}_i = \tilde{\omega}_{i-1}$ and $\hat{\omega}_i = \min\{\hat{\omega}_{i-1} + w_i, 1\}$. In addition, $x_i = \frac{\min\{w_i, 1-\hat{\omega}_{i-1}\}}{1+\hat{\omega}_i} - s_{i-1} \cdot \frac{\min\{w_i, 1-\hat{\omega}_{i-1}\}}{1+\hat{\omega}_i}$. Then we have

$$
\begin{aligned}
s_i &= s_{i-1} + x_i \\
&= \frac{\hat{\omega}_{i-1} + \tilde{\omega}_i}{1 + \hat{\omega}_{i-1}} + \frac{\min\{w_i, 1 - \hat{\omega}_{i-1}\}}{1 + \hat{\omega}_i} - \frac{\hat{\omega}_{i-1} + \tilde{\omega}_i}{1 + \hat{\omega}_{i-1}} \cdot \frac{\min\{w_i, 1 - \hat{\omega}_{i-1}\}}{1 + \hat{\omega}_i} \\
&= \frac{\min\{w_i, 1 - \hat{\omega}_{i-1}\}}{1 + \hat{\omega}_i} + \frac{\hat{\omega}_{i-1} + \tilde{\omega}_i}{1 + \hat{\omega}_{i-1}} \cdot \frac{1 + \hat{\omega}_i - \min\{w_i, 1 - \hat{\omega}_{i-1}\}}{1 + \hat{\omega}_i} \\
&= \frac{\min\{w_i, 1 - \hat{\omega}_{i-1}\}}{1 + \hat{\omega}_i} + \frac{\hat{\omega}_{i-1} + \tilde{\omega}_{i-1}}{1 + \hat{\omega}_i} \\
&= \frac{\hat{\omega}_i + \tilde{\omega}_i}{1 + \hat{\omega}_i}.
\end{aligned}
$$

where the last equality holds because from PP-a we have

$$1 + \hat{\omega}_i - \min\{w_i, 1 - \hat{\omega}_{i-1}\} = 1 + \hat{\omega}_{i-1}. \tag{25}$$

This completes the proof. □

Next, we prove that PP-a achieves at least $1/(1 + \hat{\omega}_n)$ of the offline optimal value $\text{OPT}(\mathcal{I})$, where $\hat{\omega}_n = \min\{\hat{\omega}, 1\}$ by definition.

**Lemma A.8.** *The total value of admitted items by PP-a is lower bounded*

$$\text{ALG}(\mathcal{I}) \geq \frac{\hat{\omega}_n \hat{v}}{1 + \hat{\omega}_n} + \frac{\sum_{i \in \mathcal{N}_n^o} w_i v_i}{1 + \hat{\omega}_n}. \tag{26}$$

Based on (22) and (26), we can show PP-a is $(1 + \hat{\omega}_n)$-competitive, i.e.,

$$
\begin{aligned}
\frac{\text{OPT}(\mathcal{I})}{\text{ALG}(\mathcal{I})} &\leq (1 + \hat{\omega}_n) \cdot \frac{\sum_{i \in \mathcal{N}_n^o} v_i w_i + \hat{v}\left(1 - \sum_{i \in \mathcal{N}_n^o} w_i\right)}{\sum_{i \in \mathcal{N}_n^o} w_i v_i + \hat{\omega}_n \hat{v}}, \\
&\leq 1 + \hat{\omega}_n.
\end{aligned}
\tag{27}
$$

where the last inequality holds since $\hat{\omega}_n \geq 1 - \sum_{i \in \mathcal{N}_n^o} w_i$ by definition of the critical value density. We complete the proof of Theorem 3.3 by proving Lemma A.8.

*Proof of Lemma A.8.* Let $\text{ALG}_i(\mathcal{I})$ denote the total value of PP-a after processing the $i$-th item. We show the lower bound of the online algorithm by showing the following inequality holds.

$$\text{ALG}_i(\mathcal{I}) \geq \frac{\hat{\omega}_i \hat{v}}{1 + \hat{\omega}_i} + \frac{\sum_{j \in \mathcal{N}_i^o} w_j v_j}{1 + \hat{\omega}_i}, \forall i \in [n]. \tag{28}$$

*Base Case:* $i = 1$. If item $1 \in \mathcal{N}_n^o$, then $\hat{\omega}_1 = \hat{\omega}_0 = 0$ and $x_1 = w_1$, and thus $\text{ALG}_1(\mathcal{I}) = v_1 x_1$, which satisfies (28). If item $1 \in \mathcal{N}_n^c$, then $\hat{\omega}_1 = w_1$ and $x_1 = \frac{\hat{\omega}_1}{1+\hat{\omega}_1}$. Then we have $\text{ALG}_1(\mathcal{I}) = \hat{v} x_1 = \frac{\hat{v} \hat{\omega}_1}{1+\hat{\omega}_1}$, which satisfies (28).

*Induction Step:* $i \geq 2$. Suppose (28) holds for $i - 1$, we aim to show the inequality for $i$.

*Case (i):* If item $i \in \mathcal{N}_i^o$, we have $\hat{\omega}_i = \hat{\omega}_{i-1}$ and $x_i = \frac{w_i}{1+\hat{\omega}_i}$,

$$\text{ALG}_i(\mathcal{I}) = \text{ALG}_{i-1}(\mathcal{I}) + v_i x_i,$$

$$\geq \frac{\hat{\omega}_i \hat{v}}{1+\hat{\omega}_i} + \frac{\sum_{j \in \mathcal{N}_{i-1}^o} w_j v_j}{1+\hat{\omega}_i} + \frac{w_i v_i}{1+\hat{\omega}_i},$$

$$= \frac{\hat{\omega}_i \hat{v}}{1+\hat{\omega}_i} + \frac{\sum_{j \in \mathcal{N}_i^o} w_j v_j}{1+\hat{\omega}_i}.$$

*Case (ii):* If item $i \in \mathcal{N}_i^c$, we have $\hat{\omega}_i = \min\{\hat{\omega}_{i-1} + w_i, 1\}$, then

$\text{ALG}_i(\mathcal{I}) = \text{ALG}_{i-1}(\mathcal{I}) + \hat{v} x_i,$

$$\geq \frac{\hat{\omega}_{i-1}\hat{v}}{1+\hat{\omega}_{i-1}} + \frac{\sum_{j \in \mathcal{N}_{i-1}^o} w_j v_j}{1+\hat{\omega}_{i-1}} + \frac{\hat{v}\min\{w_i, 1-\hat{\omega}_{i-1}\}}{1+\hat{\omega}_i} - s_{i-1} \cdot \frac{\hat{v}\min\{w_i, 1-\hat{\omega}_{i-1}\}}{1+\hat{\omega}_i},$$

$$= \frac{\hat{\omega}_{i-1}\hat{v}}{1+\hat{\omega}_{i-1}} \cdot \frac{1+\hat{\omega}_i - \min\{w_i, 1-\hat{\omega}_{i-1}\}}{1+\hat{\omega}_i} + \frac{\sum_{j \in \mathcal{N}_{i-1}^o} w_j v_j}{1+\hat{\omega}_{i-1}} - \frac{\hat{v}\tilde{\omega}_{i-1}}{1+\hat{\omega}_{i-1}} \cdot \frac{\min\{w_i, 1-\hat{\omega}_{i-1}\}}{1+\hat{\omega}_i} + \hat{v} \cdot \frac{\min\{w_i, 1-\hat{\omega}_{i-1}\}}{1+\hat{\omega}_i},$$

$$\geq \frac{\hat{\omega}_{i-1}\hat{v}}{1+\hat{\omega}_i} + \frac{\sum_{j \in \mathcal{N}_{i-1}^o} w_j v_j}{1+\hat{\omega}_i} + \frac{\hat{v}\min\{w_i, 1-\hat{\omega}_{i-1}\}}{1+\hat{\omega}_i},$$

$$= \frac{\hat{\omega}_i \hat{v}}{1+\hat{\omega}_i} + \frac{\sum_{j \in \mathcal{N}_i^o} w_j v_j}{1+\hat{\omega}_i}.$$

where the first inequality is obtained by using the induction hypothesis and substituting $x_i = \frac{\min\{w_i, 1-\hat{\omega}_{i-1}\}}{1+\hat{\omega}_i} - s_{i-1}\frac{\min\{w_i, 1-\hat{\omega}_{i-1}\}}{1+\hat{\omega}_i}$, the second equality is obtained by substituting $s_{i-1} = \frac{\hat{\omega}_{i-1}+\tilde{\omega}_{i-1}}{1+\hat{\omega}_{i-1}}$ from (23), and the last inequality holds because $\sum_{j \in \mathcal{N}_{i-1}^o} w_j v_j \geq \hat{v}\hat{\omega}_{i-1}$ and $1+\hat{\omega}_i - \min\{w_i, 1-\hat{\omega}_{i-1}\} = 1+\hat{\omega}_{i-1}$. $\square$

### A.6.7. PROOF OF THEOREM 3.4

*Proof.* First, we analyze the feasibility of the solution – we show that $\sum_{i=1}^n x_i \leq 1$.

$$\sum_{i=1}^n x_i = \left( \sum_{v_i < \ell} x_i \right) + \left( \sum_{v_i \in [\ell, u]} x_i \right) + \left( \sum_{v_i > u} x_i \right). \tag{29}$$

By substituting sub-algorithm selections, we have the following:

$$\sum_{i=1}^n x_i = 0 + \left( \sum_{v_i \in [\ell, u]} \frac{\alpha}{\alpha+1} \cdot x_i^A \right) + \left( \sum_{v_i > u} \frac{1}{\alpha+1} \cdot w_i \right). \tag{30}$$

Using the fact that the sub-algorithm is also a feasible algorithm, we can say that:

$$\left( \sum_{v_i \in [\ell, u]} \frac{\alpha}{\alpha+1} \cdot x_i^A \right) = \frac{\alpha}{\alpha+1} \cdot \left( \sum_{v_i \in [\ell, u]} x_i^A \right) \leq \frac{\alpha}{\alpha+1} \cdot 1. \tag{31}$$

Also, from (4), we know that the optimal solution will select every $w_i$ with $v_i > \hat{v}$, since $\hat{v} \in [\ell, u]$. We can say that $w_i$ for which $v_i > u$ implies $v_i > \hat{v}$. Moreover, (5) is a feasible result, we know that $\sum_{\hat{v} < v_i} w_i \leq 1$. So, we claim that:

$$\left( \sum_{v_i > u} \frac{1}{\alpha+1} \cdot w_i \right) = \frac{1}{\alpha+1} \cdot \left( \sum_{v_i > u} w_i \right) \leq \frac{1}{\alpha+1} \cdot \left( \sum_{\hat{v} < v_i} w_i \right) \leq \frac{1}{\alpha+1} \cdot 1. \tag{32}$$

Substituting (32) and (31) into (30), we obtain:

$$\sum_{i=1}^{n} x_i = 0 + \left( \sum_{v_i \in [\ell,u]} \frac{\alpha}{\alpha+1} \cdot x_i^A \right) + \left( \sum_{v_i > u} \frac{1}{\alpha+1} \cdot w_i \right) \leq \frac{\alpha}{\alpha+1} + \frac{1}{\alpha+1} = 1. \tag{33}$$

which completes the proof that the solutions are feasible.

We proceed to prove that the algorithm achieves a competitive ratio of $\alpha + 1$ (given sub-algorithm ZCL, which has a competitive ratio $\alpha$). The profit can be calculated based on $x_i$ decisions, using (30):

$$\text{ALG}(\mathcal{I}) = \left( \sum_{v_i \in [\ell,u]} \frac{\alpha}{\alpha+1} \cdot x_i^A \cdot v_i \right) + \left( \sum_{v_i > u} \frac{1}{\alpha+1} \cdot w_i \cdot v_i \right). \tag{34}$$

For the first sum, we know that Algorithm ZCL guarantees $\alpha$-competitiveness, which we show as follows:

$$\left( \sum_{v_i \in [\ell,u]} x_i^A \cdot v_i \right) \times \alpha \geq \text{OPT}(\mathcal{I}_0). \tag{35}$$

where $\mathcal{I}_0$ is all items in $\mathcal{I}$ such that $v_i \in [\ell, u]$. Also, we claim that:

$$\text{OPT}(\mathcal{I}_0) = \sum_{u > v_i' > \ell, i < p'} w_i' \cdot v_i' + \left( 1 - \sum_{u > v_i' > \ell, i < p'} w_i' \right) \cdot v_p^{\mathcal{I}_0}. \tag{36}$$

Where $v_p^{\mathcal{I}_0}$ represents the critical value for $\mathcal{I}_0$. We argue that $v_p^{\mathcal{I}_0} \leq \hat{v}$. If we denote $\hat{v}$ as the $p$th item in the sorted list, as defined in (5), and $v_p^{\mathcal{I}_0}$ as the $p'$th item in the sorted list of the instance $\mathcal{I}$, we assert that $p \leq p'$. The rationale behind this assertion is rooted in the definitions. Specifically, $p$ is defined as the largest number for which $\sum_{i=1}^{p-1} w_i' < 1$. Now, let us assume $k$ is the first item with a value less than or equal to $u$. By definition, $p'$ represents the largest number for which $\sum_{i=k}^{p'-1} w_i' < 1$. If we were to assume that $p > p'$, this would contradict the definition of $p'$ as the largest number, because changing $p'$ to $p$ would yield a sum less than one, but we increased from the $p'$ to another larger number. Consequently, it is not valid to claim that $p > p'$; instead, we conclude that $p \leq p'$. This implies $v_p^{\mathcal{I}_0} \leq \hat{v}$.

Using this observation, we can now compare $\text{OPT}(\mathcal{I})$ and $\text{OPT}(\mathcal{I}_0)$:

$$\begin{aligned}
\text{OPT}(\mathcal{I}) &= \sum_{i=1}^{p-1} w_i' v_i' + \left( 1 - \sum_{i=1}^{p-1} w_i' \right) \hat{v}, \\
&= \sum_{v_i > u} w_i' v_i' + \sum_{i=k}^{p-1} w_i' v_i' + \left( 1 - \sum_{i=1}^{p-1} w_i' \right) \hat{v}, \\
&\leq \sum_{v_i > u} w_i' v_i' + \sum_{i=k}^{p-1} w_i' v_i' + \sum_{i=p}^{p'-1} w_i' v_i' + \left( 1 - \sum_{i=k}^{p'-1} w_i' \right) v_p^{\mathcal{I}_0}, \\
&= \sum_{v_i > u} w_i v_i + \text{OPT}(\mathcal{I}_0).
\end{aligned} \tag{37}$$

where (37) holds because if $p = p'$, then we have $\hat{v} = v_p^{\mathcal{I}_0}$, implying that the last sum of $\text{OPT}(\mathcal{I})$ (which is $\left( 1 - \sum_{i=1}^{p-1} w_i' \right) \hat{v}$), is equal to $(\sum_{i=k}^{p'-1} w_i') v_p^{\mathcal{I}_0}$.

On the other hand, if $p < p'$, the last part of $\text{OPT}(\mathcal{I})$ can be bounded by $w_p' v_p'$, which is subsumed within $\sum_{i=p}^{p'-1} w_i' v_i'$. This follows from the fact that $1 - \sum_{i=1}^{p-1} w_i' < w_p'$ due to the definition of a feasible answer.

Using (35) and (37), we claim that:

$$\left( \sum_{v_i \in [\ell, u]} x_i^A \cdot v_i \right) \geq \frac{1}{\alpha} \cdot (\text{OPT}(\mathcal{I}) - \sum_{v_i > u} w_i v_i). \tag{38}$$

Now, let us combine this with other parts in (34):

$$\begin{aligned}
\text{ALG}(\mathcal{I}) &= \left( \sum_{v_i \in [\ell, u]} \frac{\alpha}{\alpha + 1} \cdot x_i^A \cdot v_i \right) + \left( \sum_{v_i > u} \frac{1}{\alpha + 1} \cdot w_i \cdot v_i \right), \\
&\geq \left( \frac{\alpha}{\alpha + 1} \cdot \frac{1}{\alpha} \left( \text{OPT}(\mathcal{I}) - \sum_{v_i > u} w_i v_i \right) \right) + \left( \frac{1}{\alpha + 1} \cdot \sum_{v_i > u} w_i \cdot v_i \right), \\
&= \frac{1}{\alpha + 1} \text{OPT}(\mathcal{I}).
\end{aligned} \tag{39}$$

Thus, we conclude that `IPA` using `ZCL` as the robust sub-algorithm is $\alpha + 1$ competitive. $\square$

### A.6.8. PROOF OF LEMMA A.5

To show this lower bound, we first construct a family of special instances and then show that the achievable consistency-robustness when given a prediction of the critical value is lower bounded under the constructed special instances. Assume that item weights are infinitesimally small. It is known that difficult instances for `OKP` occur when items arrive at the algorithm in a non-decreasing order of value density (Zhou et al., 2008; Sun et al., 2021b). We now formalize such a family of instances $\{\mathcal{I}_x\}_{x \in [L,U]}$, where $\mathcal{I}_x$ is called an $x$-continuously non-decreasing instance.

**Definition A.9.** Let $N, m \in \mathbb{N}$ be sufficiently large, and $\delta := (U - L)/N$. For $x \in [L, U]$, an instance $\mathcal{I}_x \in \Omega$ is $x$-continuously non-decreasing if it consists of $N_x := \lceil (x - L)/\delta \rceil + 1$ batches of items and the $i$-th batch ($i \in [N_x]$) contains $m$ items with value density $L + (i - 1)\delta$ and weight $1/m$.

Note that $\mathcal{I}_L$ is simply a stream of $m$ items, each with weight $1/m$ and value density $L$. See Fig. A1 for an illustration of an $x$-continuously non-decreasing instance.

$$\underbrace{\substack{v=L/m \\ w=1/m}, \cdots, \substack{v=L/m \\ w=1/m}}_{\text{Batch 0 with } m \text{ items}}, \quad \underbrace{\substack{v=(L+\delta)/m \\ w=1/m}, \cdots, \substack{v=(L+\delta)/m \\ w=1/m}}_{\text{Batch 1 with } m \text{ items}}, \quad \cdots, \quad \underbrace{\substack{v=x/m \\ w=1/m}, \cdots, \substack{v=x/m \\ w=1/m}}_{\text{Batch } N_x \text{ with } m \text{ items}}$$

*Figure A1.* $\mathcal{I}_x$ consists of $N_x$ batches of items, arriving in increasing order of value density.

We will operate with two types of $x$-increasing instances as follows. Let $\mathcal{I}_{x,x} = \mathcal{I}_x$ as defined in Def. A.6.8. Furthermore, let $\mathcal{I}_{x,U}$ denote a sequence defined as $\mathcal{I}_{x,U} = \mathcal{I}_x; \{U \times (1/\epsilon - 1)\}$. In other words, we append $(1/\epsilon - 1)$ items with value density $U$ to the end of the sequence $\mathcal{I}_x$. (this is the worst-case for consistency, observing that $\hat{v} = x$)

Suppose a learning-augmented algorithm `ALG` is $\gamma$-robust and $\eta$-consistent. Let $g(x) : [L, U] \to [0, 1]$ denote an arbitrary *acceptance function*, which fully parameterizes the decisions made by `ALG` under the special instances. $g(x)$ gives the final knapsack utilization (total weight of accepted items) under the instance $\mathcal{I}_x$. Note that for small $\delta$, processing $\mathcal{I}_{x+\delta}$ is equivalent to first processing $\mathcal{I}_x$, and then processing $m$ identical items, each with weight $\frac{1}{m}$ and value density $x + \delta$. Since this function $g(\cdot)$ is unidirectional (item acceptances are irrevocable) and deterministic, we must have $g(x + \delta) \geq g(x)$, i.e., $g(x)$ is non-decreasing in $[L, U]$. Once a batch of items with maximum value density $U$ arrives, the rest of the capacity should be used, i.e., $g(U) = 1$. `ALG` is $\gamma$-robust. Observe that in order to be $\gamma$-competitive under the instance $\mathcal{I}_L$, we must have that $g(L) \geq \frac{1}{\gamma}$.

Furthermore, under the instance $\mathcal{I}_x$, the online algorithm with acceptance function $g$ obtains a value of $\text{ALG}(\mathcal{I}_x) = g(L)L + \int_L^x u \, dg(u)$, where $u \, dg(u)$ is the value obtained by accepting items with value density $u$ and weight $dg(u)$. Under the instance $\mathcal{I}_x$, the offline optimal solution obtains a total value of $\text{OPT}(\mathcal{I}_x) = x$. Therefore, any $\gamma$-robust online algorithm must satisfy:

$$\text{ALG}(\mathcal{I}_x) = g(L)L + \int_L^x u \, dg(u) \geq \frac{x}{\gamma}, \quad \forall x \in [L, U].$$

By integral by parts and Grönwall's Inequality (Theorem 1, p. 356, in (Mitrinovic et al., 1991)), a necessary condition for the competitive constraint above to hold is the following:

$$g(x) \geq \frac{1}{\gamma} + \frac{1}{x} \int_L^x g(u) du \geq \frac{1}{\gamma} \left[ 1 + \ln \left( \frac{x}{L} \right) \right]. \tag{40}$$

Assume that a learning-augmented algorithm ALG receives a prediction $\hat{v}$. If the prediction is correct, we know that the items with value densities strictly greater than $\hat{v}$ have total weight less than 1. If the actual best value density is $U$, ALG must satisfy $\text{ALG}(\mathcal{I}_{\hat{v},U}) \geq \text{OPT}(\mathcal{I}_{\hat{v},U})/\eta$. Note that $g(U) = 1$ by the structure of the problem. This gives

$$g(U)U - \int_L^U g(u) du \geq \frac{U}{\eta}. \tag{41}$$

Furthermore, for ALG to be $\eta$-consistent on an instance where $\hat{v} = x$, recall that $\mathcal{I}_{x,x}$ denotes an instance where prices continually increase up to $x$, and $g(x)$ denotes the fraction of knapsack capacity filled with items of value density $\leq x$. By the definition of consistency, it follows that $g(x)$ must satisfy $g(x) \geq \frac{1}{\eta}$. Combining the above condition with the robustness condition on $g(x)$, an $\eta$-consistent and $\gamma$-robust algorithm must have $g(x) \geq \max\{\frac{1}{\gamma}[1 + \ln(\frac{x}{L})], 1/\eta\}$. Thus, we have the following:

$$\max \left\{ \int_L^U \frac{1}{\eta} dx, \; \frac{1}{\gamma} \int_L^U \left[ 1 + \ln \left( \frac{x}{L} \right) \right] \right\} dx \leq \int_L^U g(u) du \leq U - \frac{U}{\eta}. \tag{42}$$

where the last inequality is based on (41).

Then the optimal $\eta$ is obtained when the inequality is binding, which gives:

$$\frac{1}{\gamma} \int_L^U \left[ 1 + \ln \left( \frac{x}{L} \right) \right] dx = U - \frac{U}{\eta},$$
$$1 - \frac{1}{\gamma} \ln \left( \frac{U}{L} \right) = \frac{1}{\eta},$$
$$\frac{1}{1 - \frac{1}{\gamma} \ln \left( \frac{U}{L} \right)} \leq \eta.$$

$$\int_L^U \frac{1}{\eta} dx = U - \frac{U}{\eta},$$
$$\frac{U - L}{\eta} = U - \frac{U}{\eta},$$
$$2 - \frac{L}{U} \leq \eta.$$

Combining the above two cases completes the lemma.

### A.6.9. PROOF OF COROLLARY 3.5

Recall that ZCL is $(\ln(U/L) + 1)$-competitive. Letting $U = u$ and $L = \ell$, we have that $(\alpha + 1) = 2 + \ln(u/\ell)$. As $\ell$ and $u$ approach each other, $\ln(u/\ell)$ approaches $0$ – in the case of $\ell = u$, this recovers the 2-competitive result of PP-b (Algorithm 5).

### A.6.10. PROOF OF THEOREM 5.2

We first prove that given the algorithm Fr2Int is feasible, it is $\gamma \cdot \frac{1+\delta}{1-\epsilon(\lceil \log_{(1+\delta)} U/L \rceil + 1)}$ competitive, and then prove its feasibility. Let $A_i[j]$ and $R_i[j]$ denote the cumulative total value of items in range $j$ (where $j = 0, \ldots, \lceil \log_{(1+\delta)} U/L \rceil$) after processing item $i$. We claim the following inequality holds for all $i = 0, \ldots, n$ and $j = 0, \ldots, \lceil \log_{(1+\delta)} U/L \rceil$:

$$A_i[j] \geq R_i[j] \cdot \frac{1 - \epsilon(\lceil \log_{(1+\delta)} U/L \rceil + 1)}{1 + \delta}. \tag{43}$$

We prove this inequality by induction. Initially, $A_0[j] = R_0[j] = 0$, and thus the (43) holds. Suppose (43) holds for $i - 1$, we show that it also holds for $i$ in the following two cases.

*Case (i):* if $A_{i-1}[j] \geq R_i[j] \cdot \frac{1 - \epsilon(\lceil \log_{(1+\delta)} U/L \rceil + 1)}{1+\delta}$, where $R_i[j] = R_{i-1}[j] + v_i w_i \tilde{x}_i$, then we have

$$A_i[j] = A_{i-1}[j] \geq R_i[j] \cdot \frac{1 - \epsilon(\lceil \log_{(1+\delta)} U/L \rceil + 1)}{1 + \delta}.$$

*Case (ii):* if $A_{i-1}[j] < R_i[j] \cdot \frac{1 - \epsilon(\lceil \log_{(1+\delta)} U/L \rceil + 1)}{1+\delta}$, then

$$A_i[j] = A_{i-1}[j] + w_i v_i \geq A_{i-1}[j] + w_i v_i \tilde{x}_i,$$

$$\geq R_{i-1}[j] \cdot \frac{1 - \epsilon(\lceil \log_{(1+\delta)} U/L \rceil + 1)}{1+\delta} + w_i v_i \tilde{x}_i,$$

$$\geq R_i[j] \cdot \frac{1 - \epsilon(\lceil \log_{(1+\delta)} U/L \rceil + 1)}{1+\delta}.$$

where the first inequality is using the induction hypothesis and the second inequality is because the factor $\frac{1 - \epsilon(\lceil \log_{(1+\delta)} U/L \rceil + 1)}{1+\delta} \leq 1$.

With (43), we further have

$$\sum_j A_n[j] \geq \sum_j R_n[j] \cdot \frac{1 - \epsilon(\lceil \log_{(1+\delta)} U/L \rceil + 1)}{1+\delta} \geq \frac{\text{OPT}}{\gamma} \cdot \frac{1 - \epsilon(\lceil \log_{(1+\delta)} U/L \rceil + 1)}{1+\delta}. \tag{44}$$

where the last inequality holds since the fractional algorithm is $\gamma$-competitive and the offline optimum of the fractional problem is no smaller than that of the integral problem.

In the following, we prove that the online solution of `Fr2Int` is feasible. Define $w(A_n[j])$ and $w(R_n[j])$ as the weight of all items in range $j$ of `Fr2Int` and `ALG` after processing the last item $n$, respectively. Based on the value partitioning in definition 5.1, the value density of items in range $j$ is lower bounded by $L \cdot (1+\delta)^j$ and upper bounded by $L \cdot (1+\delta)^{j+1}$. Thus, we have:

$$w(A_n[j]) \cdot L \cdot (1+\delta)^j \leq A_n[j], \text{ and, } R_n[j] \leq w(R_n[j]) \cdot L \cdot (1+\delta)^{j+1}.$$

Let $i_j$ denote the last item admitted by `Fr2Int` in the range $j$. Then we have

$$A_{i_j-1}[j] < R_{i_j}[j] \cdot \frac{(1 - \epsilon(\lceil \log_{(1+\delta)} U/L \rceil + 1))}{(1+\delta)},$$

$$\leq w(R_{i_j}[j]) \cdot L \cdot (1+\delta)^{j+1} \cdot \frac{(1 - \epsilon(\lceil \log_{(1+\delta)} U/L \rceil + 1))}{(1+\delta)}.$$

Since $A_{i_j-1}[j] \geq w(A_{i_j-1}[j]) \cdot L \cdot (1+\delta)^j$, combining with above equation gives

$$w(A_{i_j-1}[j]) \leq w(R_{i_j}[j]) \cdot (1 - \epsilon(\lceil \log_{(1+\delta)} U/L \rceil + 1)). \tag{45}$$

Then we further have

$$w(A_{i_j}[j]) \leq \epsilon + w(A_{i_j-1}[j]) \leq \epsilon + w(R_{i_j}[j]) \cdot (1 - \epsilon(\lceil \log_{(1+\delta)} U/L \rceil + 1)). \tag{46}$$

Summing the weights over all ranges gives

$$\sum_j w(A_{i_j}[j]) \leq \epsilon \cdot (\lceil \log_{(1+\delta)} U/L \rceil + 1) + \sum_j w(R_{i_j}[j]) \cdot (1 - \epsilon(\lceil \log_{(1+\delta)} U/L \rceil + 1)) \leq 1. \tag{47}$$

where the last inequality holds since $\sum_j w(R_{i_j}[j]) \leq 1$. Thus, the solution of `Fr2Int` is feasible.

### A.7. Additional Numerical Experiments

In this section, we discuss other sets of experiments and report additional experimental results of the proposed algorithms' performance.

#### A.7.1. OTHER EXPERIMENTAL RESULTS SETUP

We talked about part of the first set of experiments in 6.We should add that an `MIX` instance is created with uniformly random prediction intervals, where each instance makes a correct prediction with a fixed probability of $1 - \delta$ and chooses a

wrong prediction otherwise. For `IPA`, we used a uniformly random interval around the correct critical value since it is a trusted prediction.

The second set of experiments compares against the existing learning-augmented `OKP` algorithm that uses frequency predictions (SENTINEL) (Im et al., 2021) using the synthetic data set from the original paper. This data set constructs a sequence of items with a small weight of 0.0001 and values between 1 to 100, where each value appears (on average) $100(1 + \delta/2)$ times in sequence. It has a lower bound and an upper bound with a fixed ratio $\text{up}/\text{low} = (1 + \delta)$, where $\delta$ is also a parameter showing the amount of prediction error as used in (Im et al., 2021).

In the third set of experiments, we use a historical data set of Bitcoin prices (Kottarathil, 2018) in 2017-2019 to evaluate algorithms under realistic data. Each instance is constructed by randomly sampling 10,000 prices from one month of the data, setting all item sizes equal to 0.001. We used the same notation of $\delta$ from (Im et al., 2021) to show the the accuracy of this experiment.

Finally, in the fourth set of experiments we follow related work (Sun et al., 2022) and construct instances based on Google cluster traces (Reiss et al., 2012). This data set records information about compute jobs on a cluster, with many short jobs and few long jobs. To construct values based on these job durations, we first scale each duration by a random number between 1 and 250, followed by a resource scale factor randomly chosen from the set $\{0.01, 0.03, 0.05\}$. Each instance includes 10,000 items generated as above, each with size 0.001.

### A.7.2. EXPERIMENT RESULTS

To convey the benefits of succinct predictions, Fig. A2 presents results on the synthetic data set and prediction method used in the original paper presenting the SENTINEL algorithm that uses frequency predictions. We compare `PP-a`, `Fr2Int-PP-a`, and `ZCL` against SENTINEL. As $\delta$(error parameter introduced by (Im et al., 2021)) grows, the number of items increases, and the prediction becomes worse. Here, it is worth mentioning that due to the way that they made their dataset, $\delta$ is also a parameter for a range of values. which make `ZCL` to act worse as $\delta$ grows because it grows $\text{up}/\text{low} = (1 + \delta)$. For `PP-a`, we use the average of the upper and lower bounds in the frequency prediction to derive a single number indicating the average critical value $\hat{v}$ And we use this number as the prediction to `PP-a`. As shown in Fig. A2, a single prediction, in practice, outperforms complex predictions.

In Fig. A3, we use a real data set of Bitcoin prices for 2017-19. Each month consists of 10,000 prices. To derive frequency predictions for SENTINEL, we generate two random numbers, $a$ and $b$, between 1 and $1 + \delta$. Each upper bound is set to $a \times s$, where $s$ is the true frequency of a given price, and the lower bounds are set to $s/b$. As in the previous experiment, our critical value prediction is derived from this frequency prediction. While `PP-a`, `Fr2Int-PP-a`, and SENTINEL all perform well, achieving competitive ratios close to 1, our algorithms outperform SENTINEL on average.

Fig. A4 plots a similar comparison for the same algorithms on the real data set of Google cluster traces. Frequency predictions for SENTINEL are generated using the same technique described above. In this experiment, we find that `PP-a` and `Fr2Int-PP-a` significantly outperform SENTINEL – this is likely the case due to the underlying distribution of values (i.e., job duration) in the cluster traces, which make the frequencies (and hence the predictions) of values less robust to error.

It is worth mentioning that for evaluating `MIX` performance we presented probabilistic input that has the probability of $1 - \delta$ for the correctness of the prediction and the probability of $\delta$ for prediction being wrong. In Figure A5(a), we evaluate the performance of `IPA` for different interval prediction widths, given as a percentage (higher is worse). As shown in Theorem 3.4, we find that tighter prediction intervals yield better empirical performance. Furthermore, all `IPA` algorithms outperform the baseline robust `ZCL` algorithm. In Figure A5(b), we evaluate the performance of `MIX` for *untrusted predictions*. We test regimes where $1 - \delta$ (probability of correct prediction) is 10%, 20%, and 50%; we fix $\lambda = 0.9$, and the interval is 20% of $[L, U]$. We find that the performance of `MIX` smoothly degrades, and even bad predictions result in an algorithm that outperforms the robust baseline `ZCL`. Finally, in Figure A5(c), we show a similar result for `MIX`– we fix $\delta = 50\%$ and vary the trust parameter $\lambda \in \{0.3, 0.5, 0.9\}$, showing that when predictions are sufficiently good, `MIX` performs better when the predictions are trusted more (i.e., increasing $\lambda$).

In Figure A6(a), we evaluate the performance of `MIX` for different values of $1 - \delta$. This plot is a CDF plot version of Figure A5(b), which illustrates how increasing the probability of correct predictions (corresponding to empirically more accurate machine-learned predictions), we consistently achieve a better competitive ratio, both on average and in the worst case.

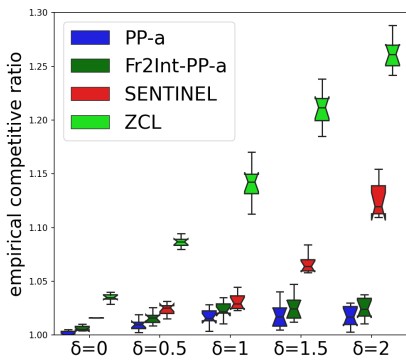

*Figure A2.* Empirical competitive ratios of `PP-a`, `Fr2Int-PP-a`, SENTINEL, and ZCL on synthetic data from original SENTINEL paper (Im et al., 2021).

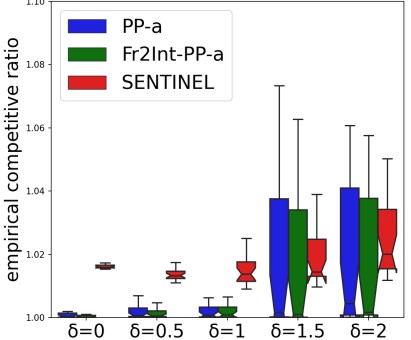

*Figure A3.* Empirical competitive ratios of `PP-a`, `Fr2Int-PP-a`, and SENTINEL in experiments on real Bitcoin data (Kottarathil, 2018).

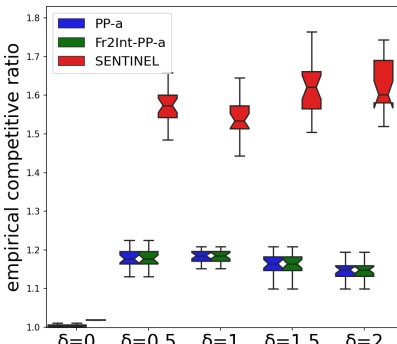

*Figure A4.* Empirical competitive ratios of `PP-a`, `Fr2Int-PP-a`, and SENTINEL in experiments on Google-Traces data.

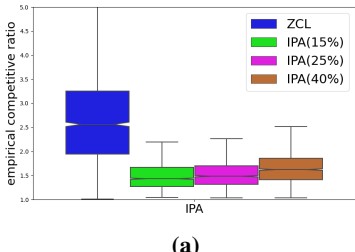

(a)

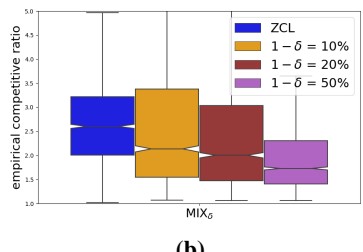

(b)

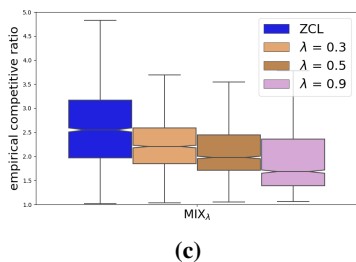

(c)

*Figure A5.* Interval-prediction-based algorithms with different interval sizes, probabilities $\delta$, and trust parameters $\lambda$: **(a)** Competitive ratios of three interval widths in `IPA` against baseline ZCL; **(b)** Competitive ratio of three probability $\delta$ values in `MIX` against baseline ZCL. $\lambda = 0.9$ and interval 20%; and **(c)** Competitive ratio of three $\lambda$ values in `MIX` against ZCL. $\delta = 50\%$ and interval 20%.

In Figure A6(b), we examine the trust parameter $\lambda$ with values $0.3, 0.5, 0.9$ for `MIX`. This figure represents a CDF plot of Figure A5(c) and shows that as the algorithm increases its trust in predictions (i.e. by increasing $\lambda$), the average competitive ratio improves. However, the worst-case competitive ratio (represented in this plot by the *tail* of the CDF) will deteriorate faster when placing more trust in predictions.

Similarly, Figure A6(c) demonstrates that the width of the prediction interval (tested as 10%, 30%, and 50% of the "width" of the interval $[L, U]$) also has an slight effect on the average and worst-case competitive ratio for `MIX`– namely, tighter prediction intervals yield better empirical performance, which aligns with our expectations.

In Figure A7, which is a CDF plot of Figure A5(a), we evaluate the empirical competitive ratio of `IPA` for various interval widths, represented as a percentage (where higher values indicate worse performance). We test intervals that are 15%, 25%, and 40% as "wide" as $[L, U]$. Intuitively, reducing the interval size results in a better competitive ratio, because the bounds on the value of $\hat{v}$ are tighter.

Figure A8 is a box plot version of Figure A5(c), illustrating that `MIX`'s average-case performance is not significantly impacted by the width of the predicted interval, although tighter intervals are still intuitively better.

Finally, in Figure A13, we vary the value of $\hat{\omega}$ in four CDF plots, one for each tested value. In contrast to `PP-n`, `PP-b`, and ZCL, these results show that the performance of `PP-a` substantially improves with smaller values of $\hat{\omega}$, confirming the results in Theorem 3.3, which establish that `PP-a`'s competitive ratio depends on $\hat{\omega}$. These figures correspond to the CDF plot of Figure 2(c). Another interesting observation is that as $\hat{\omega}$ decreases, `PP-a`'s empirical performance worsens, but for high $\hat{\omega}$ values, it improves. This occurs because `PP-a` is designed to target 2-competitiveness, but performs well when selecting more items in $\hat{v}$ than the optimal solution. This occurs when $\hat{\omega}$ is high.

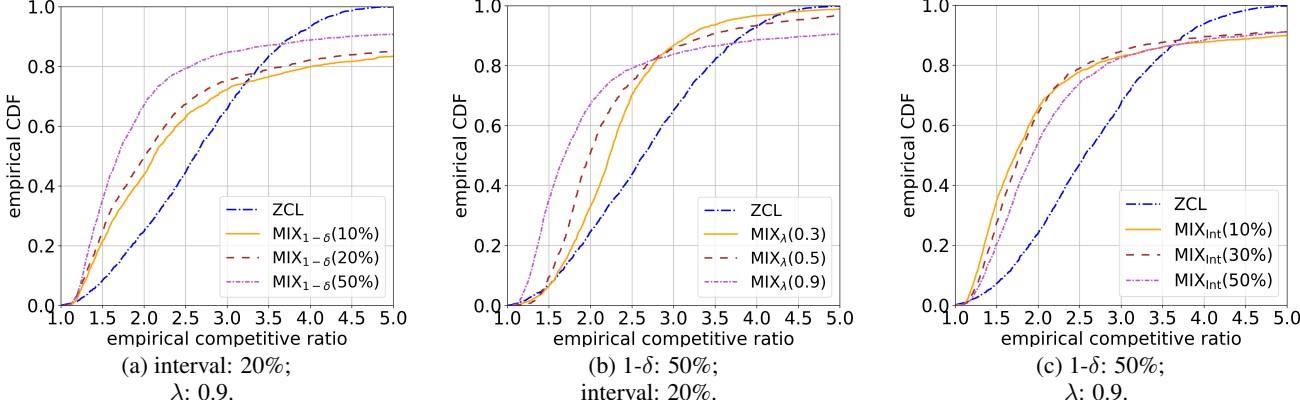

*Figure A6.* Performance of the meta-algorithm (`MIX`) when provided with interval predictions, with varying parameters (probability of correct prediction $(1 - \delta)$, trust parameter $\lambda$, and interval size) against the robust threshold-based algorithm (`ZCL`).

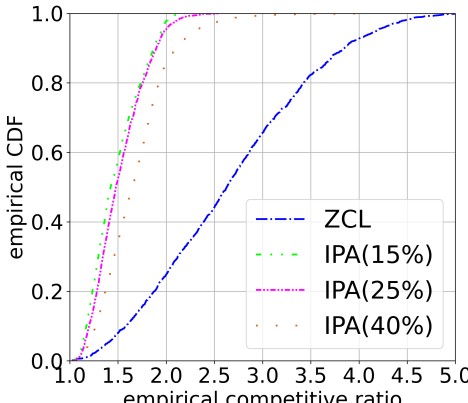

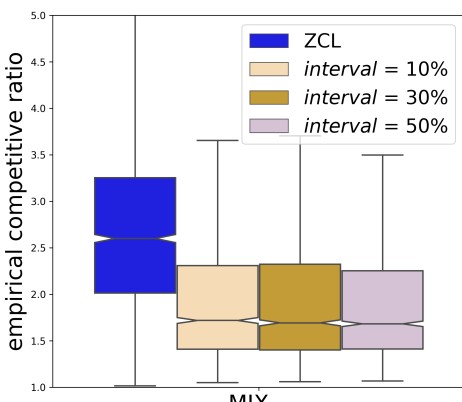

*Figure A7.* The performance of interval-prediction-based algorithm (`IPA`) with three intervals against online threshold-based algorithm (`ZCL`).

*Figure A8.* Performance of meta-algorithm (`MIX`) when provided with interval predictions versus threshold-based algorithm (`ZCL`). $\delta = 50\%, \lambda = 0.9$

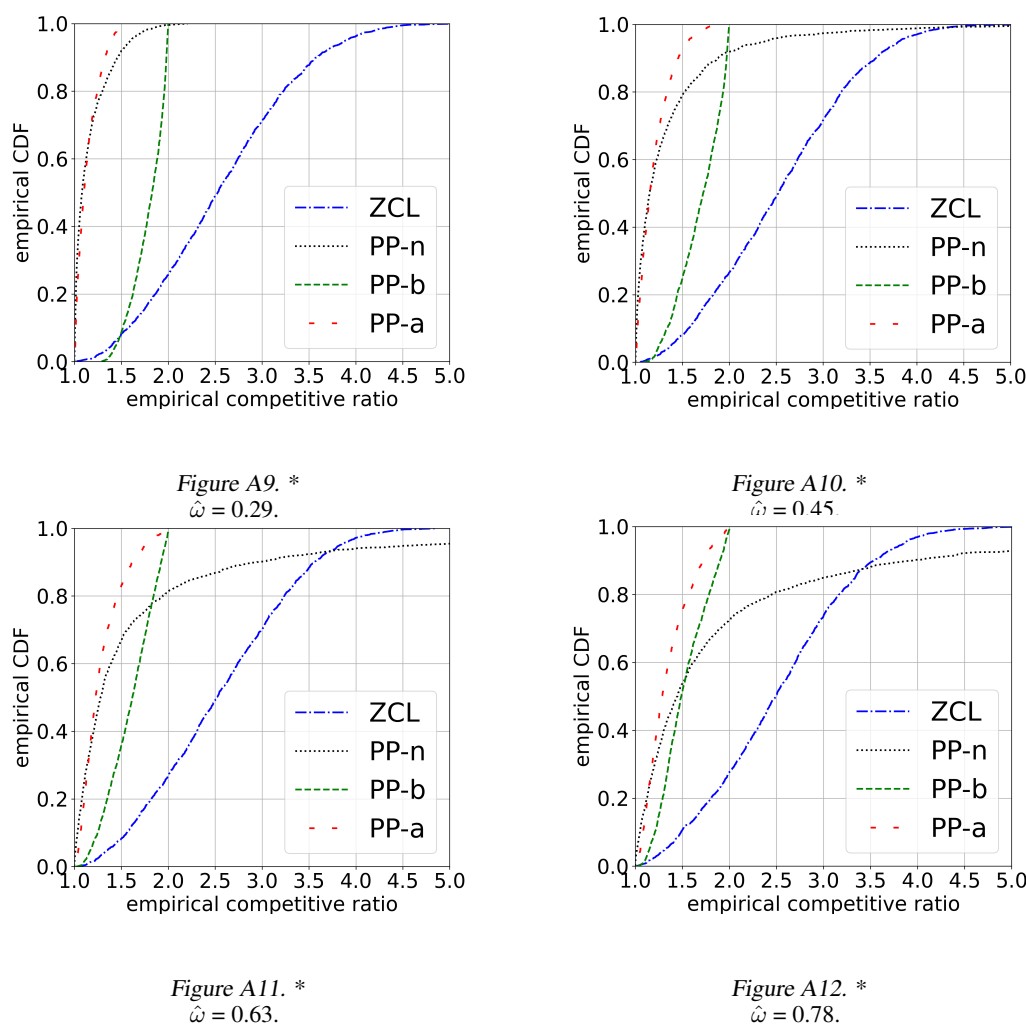

*Figure A9.* *
$\hat{\omega} = 0.29$.

*Figure A10.* *
$\hat{\omega} = 0.45$.

*Figure A11.* *
$\hat{\omega} = 0.63$.

*Figure A12.* *
$\hat{\omega} = 0.78$.

*Figure A13.* Performance comparison of naïve greedy algorithm (`PP-n`), basic 2-competitive algorithm (`PP-b`), and improved $1 + \min\{1, \hat{\omega}\}$-competitive algorithm (`PP-a`) against threshold-based algorithm (`ZCL`) with varying $\hat{\omega}$.

