# OpenReview forum: "Near-Optimal Consistency-Robustness Trade-Offs for Learning-Augmented Online Knapsack Problems"
_ICML.cc/2025/Conference — ICML 2025 poster_

### Official Review · Reviewer_jEdr · 2025-03-08

**Overall Recommendation:** 4

**Summary:**

This paper introduces learning-augmented algorithms for the online knapsack problem (OKP) that balance consistency (performance with accurate predictions) and robustness (worst-case guarantees). The authors propose algorithms leveraging succinct predictions (point or interval estimates of the critical value in the optimal solution) and provide theoretical guarantees, including matching lower bounds. They also present a fractional-to-integral conversion method and empirical validation on synthetic and real-world datasets. This leads to all proposed algorithms for OFKP work for OIKP.

**Claims And Evidence:**

The claims made in the submission are supported by clear and convincing evidence.

**Essential References Not Discussed:**

All necessary related works are discussed in the submission.

**Experimental Designs Or Analyses:**

I only made a quick pass on experimental designs, and it looks reasonable. Note that the main contribution of this paper is theoretical.

**Methods And Evaluation Criteria:**

The proposed methods make sense for the problem.

**Other Comments Or Suggestions:**

N/A

**Other Strengths And Weaknesses:**

Strengths

1.  The paper provides rigorous analyses, including near-optimal consistency-robustness trade-offs. The lower bounds (Theorems 3.1, 3.6, 4.2) tightly align with the algorithms’ guarantees, demonstrating the Pareto optimality.
2. In my view, the technical contribution reaches the bar of ICML. PP-a’s dynamic reservation strategy and IPA’s interval handling are novel. The MIX algorithm combines trusted and robust baselines, achieving near-optimal trade-offs. I also appreciate the rounding algorithm, which might be useful for another learning-augmented algorithm.
3. This work also includes the experiments. The synthetic and real-data experiments validate theoretical claims, showing robustness to prediction noise and outperforming baselines like ZCL and SENTINEL.

Weakness:

1. The proposed algorithm deeply relies on the assumption of small item weights. However, I understand that this is a standard assumption in the literature. However, it is still interesting to see algorithms without this assumption.
2. The presentation can be further improved. For example, algorithm PP-a's “prebuying” idea deserves more intuition in the main text.

**Questions For Authors:**

N/A

**Relation To Broader Scientific Literature:**

The key contribution of this paper is inspired by the previous learning-augmented online knapsack and classical knapsack algorithms. But, this paper still provides sufficient technical contribution.

**Theoretical Claims:**

I checked the correctness of some claims but didn't go through all the proofs. The proofs that I have checked are reasonable and sound.

---

> ### Author Rebuttal · Authors · 2025-04-01
>
> Thank you very much for your positive review and your constructive feedback.
>
>
> **On the small item weight assumption:** We agree that this is a standard and important assumption in the integral setting. Our lower bounds (e.g., Theorem A.1) show that it is necessary for meaningful guarantees in our setting. But we agree that exploring other settings, or making additional assumptions that allow the small weight assumption to be relaxed, would be interesting.
>
>
> **On clarifying the intuition behind “prebuying” in PP-a:** Thank you for this suggestion. We will improve the explanation of PP-a’s dynamic reservation strategy in the main text and use a small running example to provide better intuition for the “prebuying” mechanism.

---

### Official Review · Reviewer_qrRA · 2025-03-08

**Overall Recommendation:** 4

**Summary:**

This paper studies the online (integral/fractional) knapsack problem under the learning-augmented framework. The prediction is either a single value revealing the smallest unit value of items included by the optimal offline solution or an interval containing this value. When the prediction is trusted and items are fractional, the paper gives an algorithm with the optimal competitive ratio for both the value prediction and the interval prediction. When the prediction is untrusted and items are fractional, the paper gives algorithms that are both consistent and robust by combining a trusted algorithm and a robust algorithm, and it shows that the consistency-robustness trade-off of its algorithm is optimal. Moreover, the paper gives an algorithm to convert an algorithm for fractional items to an algorithm for integral items with almost the same guarantees, under the assumption that the weight of each item is small enough. Finally, the paper conducts experiments to validate the empirical performance of its algorithms.

**Claims And Evidence:**

The claims in this paper are all supported by rigorous and clear proofs.

**Essential References Not Discussed:**

To the best of my knowledge, there is no essential references missing.

**Experimental Designs Or Analyses:**

The experiments are sound and valid.

**Methods And Evaluation Criteria:**

The proposed methods make sense.

**Other Comments Or Suggestions:**

I don't have further comments.

**Other Strengths And Weaknesses:**

This paper studies an important and interesting problem closely related to the ICML community. It is also well-written, and I enjoy reading it. This paper gives fairly comprehensive results regarding the online knapsack problem under the critical-value (or interval) prediction, and the proofs are non-trivial. In particular, the prediction form considered by this paper is arguably much simplier than the predictions in prior work for the same problem, and all the technical assumptions made in the paper are reasonable and well-justified. Overall, I believe this paper clearly exceeds the bar for acceptance.

**Questions For Authors:**

I don't have further questions.

**Relation To Broader Scientific Literature:**

The paper is broadly related to the literature on learning-augmented algorithms and the online knapsack problems.

**Theoretical Claims:**

The proofs are correct to the extent that I have checked.

---

> ### Author Rebuttal · Authors · 2025-04-01
>
> Thank you for your thoughtful and positive review.
> We especially appreciate your recognition of the simplicity and practicality of our prediction models, as well as the value of extending results to the integral setting. We will continue to refine the writing and presentation to further improve clarity, especially in the introduction and technical sections.

---

### Official Review · Reviewer_jFAU · 2025-03-10

**Overall Recommendation:** 3

**Summary:**

The paper considers the online knapsack problem with predictions. In this problem, we are given a knapsack and a set of items that arrive sequentially. When each item arrives, its value and weight are revealed, and we must decide immediately and irreversibly whether to place the item in the knapsack. The goal is to maximize the total value of the selected items while ensuring that the knapsack's capacity constraint is not exceeded.

The authors propose two prediction models: in the first model, the algorithm has access to the predicted minimum item value in the offline OPT; in the second model, the algorithm has access to an interval in which the minimum item value in the offline OPT falls. The paper designs algorithms for both of them. Specifically, the authors first assume that the predictions are accurate and design semi-online algorithms for each model. They then use a simple random combination between the proposed semi-online algorithms and a classic robust algorithm to maintain robustness and consistency. The superiority of the proposed algorithms is demonstrated through theoretical lower-bound proofs and empirical performance evaluations.

## update after rebuttal
I appreciate the authors' rebuttal and will keep my original score.

**Claims And Evidence:**

Yes

**Essential References Not Discussed:**

No

**Experimental Designs Or Analyses:**

Yes

**Methods And Evaluation Criteria:**

The proposed method makes sense, and the paper provides both theoretical and empirical evaluations.

**Other Comments Or Suggestions:**

- It might be better to use Yao's minimax principle to prove Theorem 3.1, as it would make the proof look cleaner.

- The paper repeatedly claims that without either of the two assumptions, no algorithm can achieve a meaningful competitive ratio. However, this is not necessarily true. A simple randomized algorithm (e.g., accepting items larger than $\hat{v}$ with half probability and accepting only \(\hat{v}\) with half probability can achieve an expected competitive ratio of 2.

**Other Strengths And Weaknesses:**

The paper's writing could be improved. The first two sections feel somewhat disorganized, as they frequently switch between introducing the authors' work and discussing related literature. Moreover, the paper refers to an item's cost-effectiveness as its "value" rather than using "value" to denote the item's actual worth, which seems weird to me.

**Questions For Authors:**

See the weakness mentioned above.

**Relation To Broader Scientific Literature:**

The paper makes contributions to the field of online problems. However, although the authors claim that their goal is to design learning-augmented algorithms, I am inclined to view their work as primarily focused on semi-online algorithm design. The learning-augmented results are achieved via a straightforward random combination, without incorporating many advanced techniques.

**Theoretical Claims:**

I checked most of the proofs. One issue I found is that in Theorem 4.2, the lower bound is only proven for deterministic algorithms, rather than for any algorithm as claimed in the theorem.

---

> ### Author Rebuttal · Authors · 2025-04-01
>
> Thank you for your detailed and thoughtful review. You have raised some great points which will help clarify the paper.
>
> **On Theorem 4.2 and randomized algorithms:** Thank you for the careful reading – you are correct that Theorem 4.2 is currently proven only for deterministic algorithms. We will revise the paper to make this explicit in both the theorem statement and the surrounding discussion.
>
> We strongly suspect that the theorem can be extended to randomized algorithms with some small modifications, and we will attempt to do so for the camera-ready. As evidence for this, note that the ZCL algorithm [4], which achieves the optimal worst-case competitive ratio for both fractional and integral OKP under a bounded value range, is itself deterministic, and cannot be improved on by any randomized algorithm.
>
> To extend Theorem 4.2 to randomized algorithms, a first step is to modify the existing proof of Lemma A.5. Specifically, we can replace the deterministic utilization function with its expectation under a randomized algorithm—an approach aligned with the proof structure in Lemma A.6, which already reasons about expected capacity usage.
>
> Thanks again for raising this point—we believe addressing it will significantly strengthen the clarity and completeness of the paper. Some of our lower bounds do hold under both randomized and deterministic algorithms (e.g., Theorem 3.1). We will make this explicit in their statements.
>
> **On the reviewer’s suggested randomized algorithm to avoid the small weights assumption:** We appreciate the suggestion of a simple randomized strategy that mixes between accepting all items with unit value ≥ v^\hat{v} and accepting only items with unit value equal to v^\hat{v}. This algorithm does indeed achieve a 2-competitive ratio in the oblivious adversary model, and does not require a bounded weight assumption.
>
> However, under an adaptive adversary, meaning that an adversary can change the input sequence “on the fly” based on the algorithm’s past decisions, this approach can fail. E.g., the first item presented can have unit value \hat{v} and weight 1. If the algorithm does not accept it, the adversary halts the input sequence. If the algorithm does accept it, the adversary may follow with a very high-value item with weight slightly less than 1, which cannot be packed due to limited capacity.
>
> For this reason, the mentioned randomized algorithm does not contradict our lower bounds, which hold for an adaptive adversary. The adaptive adversary model is the standard model in the online algorithms literature [1,2]. However, we agree that we should add text throughout that explicitly mentions which adversarial model we are working under. We will also qualify the statement about small item weights being required by saying that this is for an adaptive adversary model only, and will note the simple randomized baseline for oblivious adversaries given by the reviewer.  Thanks again for pointing this out.
>
> **On using Yao’s Minmax for Theorem 3.1:** Thanks for the suggestion – we agree we should be able to do this and clean up the proof a bit.
>
> **On terminology "value" versus "unit value":** Thank you for pointing this out. We use “value” to refer to the unit value (i.e., value-to-weight ratio). This convention is common in the online knapsack literature. For example, it is used in prior work by [3, 5] and other papers on learning-augmented knapsack. Nonetheless, we will revise the paper to clarify our terminology up front and make our usage more precise.
>
> [1] Adam Lechowicz, Nicolas Christianson, Bo Sun, Noman Bashir, Mohammad Hajiesmaili, Adam Wierman, and Prashant Shenoy. 2025. Learning-Augmented Competitive Algorithms for Spatiotemporal Online Allocation with Deadline Constraints. Proc. ACM Meas. Anal. Comput. Syst. 9, 1, Article 8 (March 2025), 49 pages.
>
> [2] Cygan, Marek & Jeż, Łukasz. (2014). Online Knapsack Revisited. Theory of Computing Systems. 58. 10.1007/978-3-319-08001-7_13.
>
> [3] S. Im, R. Kumar, M. Montazer Qaem, and M. Purohit. Online Knapsack with Frequency Predictions. In Advances in Neural Information Processing Systems (NeurIPS), volume 34, pages 2733–2743, 2021.
>
> [4] Y. Zhou, X. Lin, and H. Zhao. Online budgeted truthful matching: A randomized primal-dual approach. Theoretical Computer Science, 2008.
>
> [5] Bo Sun, Lin Yang, Mohammad Hajiesmaili, Adam Wierman, John C. S. Lui, Don Towsley, and Danny H.K. Tsang. 2022. The Online Knapsack Problem with Departures. Proc. ACM Meas. Anal. Comput. Syst. 6, 3, Article 57 (December 2022), 32 pages.

---

### Official Review · Reviewer_6Cmk · 2025-03-16

**Overall Recommendation:** 4

**Summary:**

The paper considers the OKP problem based on succinct predictions and design learning-augmented algorithm to achieve a good trade-off between robustness and consistency. The succinct prediction model provides either a single-valued prediction or an interval prediction. The paper first consider trusted predictions and design competitive algorithms for both point prediction and interval-prediction, respectively. Next, the paper considers untrusted predictions and prove the robustness-consistency tradeoff for the algorithm MIX which linearly combines the prediction based solution and the robust algorithm ZCL. Further, the paper extends the algorithm from the fractional setting to the integral setting. The authors present a case study on both synthetic and real datasets.

**Claims And Evidence:**

Yes.

**Essential References Not Discussed:**

No.

**Experimental Designs Or Analyses:**

The experimental designs look sound.

**Methods And Evaluation Criteria:**

The paper considers competitive ratios under trusted or untrusted predictions.  This evaluation criteria is common for OKP problems. The empirical competitive ratios are evaluated in numerical experiments, which also makes sense.

**Other Comments Or Suggestions:**

N/A

**Other Strengths And Weaknesses:**

The paper consider the succinct predictions which are different from previous prediction information model.  The optimality of the robustness-consistency tradeoff is justified by proving the lower bounds.  The algorithm is also extended to the OIKP setting and the competitive ratio bound shows the effect of the rounding error $\delta$.

The weakness is that the learning-augmented design MIX is a simple linear combination of the prediction-based and prediction-free results, so there is no novelty from the  learning-augmented algorithm design.
My concern is about the robustness of MIX. Given any possible prediction, the performance of prediction-based algorithm $\hat{x}$ can be arbitrarily bad. In such case, how can we get a finite competitive ratio given a non-zero $\lambda$?

**Questions For Authors:**

My concern is about the robustness of MIX. Given any possible prediction, the performance of prediction-based algorithm $\hat{x}$ can be arbitrarily bad. In such case, how can we get a finite competitive ratio given a non-zero $\lambda$? Please let me know if I miss something.

**Relation To Broader Scientific Literature:**

The contributions of this paper are closely related to broader literature of learning-augmented algorithms for online problems.

**Theoretical Claims:**

I didn't check the detailed proofs of theoretical claims.

---

> ### Author Rebuttal · Authors · 2025-04-01
>
> Thank you for your thoughtful review and for highlighting the key question about the robustness of MIX when predictions may be arbitrarily incorrect.
>
> **On the robustness guarantee of MIX despite poor predictions:** The MIX algorithm handles untrusted predictions by combining the decisions of a robust baseline (ZCL) and a prediction-based algorithm using a parameter $\lambda \in (0,1)$. Since this is a maximization problem, the competitive ratio is defined as $\text{OPT} / \text{ALG}$. Even if the prediction-based algorithm performs poorly (e.g., contributes zero profit), the robust algorithm still guarantees a fraction of OPT.
> To make this concrete, consider a simple example: suppose the algorithm allocates half the capacity to the robust algorithm and half to the prediction-based one. If the prediction is arbitrarily wrong and the prediction-based component gets zero value, then the robust half still contributes half of its guaranteed performance. The total gain is then:
> $\text{ALG} = \frac{1}{2} \cdot \text{ALG}{\text{pred}} + \frac{1}{2} \cdot \text{ALG}{\text{robust}} = 0 + \frac{1}{2} \cdot \frac{\text{OPT}}{C} = \frac{\text{OPT}}{2C}$
>
> where C is the worst case competitive ratio achieved by $ALG_{robust}$. Thus, the competitive ratio of ALG is: $\frac{\text{OPT}}{\text{ALG}} = \frac{\text{OPT}}{\text{OPT}/(2C)} = 2C$.
>
> Since ZCL is $C = \ln(U/L) + 1$-competitive, the MIX algorithm remains $2 \cdot (\ln(U/L) + 1)$-competitive in this extreme case. This reasoning critically relies on the fact that we are solving a maximization problem, where getting zero gain from the learning-augmented part of the algorithm still results in a bounded ratio. In contrast, in minimization settings, such an approach could lead to an unbounded competitive ratio if one component incurs a large cost. We will clarify this in the final version.

---

> > ### Comment · Reviewer_6Cmk · 2025-04-05
> >
> > Thank you for addressing my questions. I have no further concerns.

---

### Decision · Program_Chairs · 2025-05-01

**Decision:**

Accept (poster)

**Comment:**

All reviewers agree that the paper makes a solid contribution to learning-augmented algorithms for online knapsack, with rigorous theoretical analysis and strong empirical validation. They praise the clarity of the proofs, the near-optimal consistency–robustness trade-offs, and the extension from fractional to integral items under reasonable assumptions.